# PRO-IP-seq tracks molecular modifications of engaged Pol II complexes at nucleotide resolution

Anniina Vihervaara [1,2] ✉, Philip Versluis[2], Samu V. Himanen[1] & John T. Lis [2] ✉

RNA Polymerase II (Pol II) is a multi-subunit complex that undergoes covalent modifications as transcription proceeds through genes and enhancers. Rate-limiting steps of transcription control Pol II recruitment, site and degree of initiation, pausing duration, productive elongation, nascent transcript processing, transcription termination, and Pol II recycling. Here, we develop Precision Run-On coupled to Immuno-Precipitation sequencing (PRO-IP-seq), which double-selects nascent RNAs and transcription complexes, and track phosphorylation of Pol II C-terminal domain (CTD) at nucleotide-resolution. We uncover precise positional control of Pol II CTD phosphorylation as transcription proceeds from the initiating nucleotide (+1 nt), through early (+18 to +30 nt) and late (+31 to +60 nt) promoter-proximal pause, and into productive elongation. Pol II CTD is predominantly unphosphorylated from initiation until the early pause-region, whereas serine-2- and serine-5-phosphorylations are preferentially deposited in the later pause-region. Upon pause-release, serine-7-phosphorylation rapidly increases and dominates over the region where Pol II assembles elongation factors and accelerates to its full elongational speed. Interestingly, tracking CTD modifications upon heat-induced transcriptional reprogramming demonstrates that Pol II with phosphorylated CTD remains paused on thousands of heat-repressed genes. These results uncover dynamic Pol II regulation at rate-limiting steps of transcription and provide a nucleotide-resolution technique for tracking composition of engaged transcription complexes.

Regulation of transcription defines cell type-specific gene expression programs and coordinates cellular responses in health and disease. In each cell type, a distinct set of genes and enhancers are transcribed by RNA Polymerase II (Pol II) complexes. To complete a full cycle of transcription, chromatin and Pol II must undergo a series of steps: (i) chromatin opening, (ii) assembly of the Pre-Initiation Complex (PIC), (iii) initiation of transcription, (iv) promoter-proximal Pol II pausing, (v) promoter-proximal Pol II pause-release, (vi) productive elongation, (vii) co-transcriptional RNA processing, (viii) transcript cleavage, (ix)

transcription termination, and (x) recycling of Pol II (reviewed in ref. 1). These steps of transcription can be modulated by the action of transcription and elongation factors, chromatin remodelers and other co-factors, architectural proteins, and RNA-binding proteins (reviewed in refs. 2,3). Additionally, transcription-regulating RNA-species and their associated RNA-binding factors can form 'RNA clouds' over clusters of active genes and their distal regulatory elements (reviewed in[4]).

As Pol II proceeds through the transcription cycle, its phosphorylation state changes dramatically. In particular, the C-terminal domain

[1]KTH Royal Institute of Technology, Department of Gene Technology, Science for Life Laboratory, Stockholm, Sweden. [2]Department of Molecular Biology and Genetics, Cornell University, Ithaca, NY 14853, USA. ✉e-mail: viher@kth.se; jtl10@cornell.edu

(CTD) of the largest subunit in Pol II consists of $Y_1S_2P_3T_4S_5P_6S_7$ repeats that undergo post-translational modifications (PTMs) and form a protean platform for regulatory interactions (reviewed in refs. 5,6). The initiating Pol II is mainly unphosphorylated, but CDK7-mediated phosphorylation of serine-5 of the CTD disrupts Pol II interaction with TATA box Binding Protein (TBP) and the Mediator, allowing Pol II to escape the PIC[7–11]. After initiation, Pol II rapidly synthetizes 20–60 nucleotides (nts) of RNA, and pauses[12–16]. The promoter-proximal Pol II pause has emerged as a major rate-limiting step[17–20] via which genome-wide transcription programs can be coordinated[21–26]. However, the CTD-modifications and regulatory signals along the promoter-proximal pause-region remain enigmatic. Multiple laboratories have shown that the release of Pol II from the promoter-proximal pause requires Positive Transcription Elongation Factor b (P-TEFb), the CDK9 subunit of which phosphorylates Negative Elongation Factor (NELF), transcription elongation factor SPT5 and serine-2 of Pol II CTD (reviewed in[27,28]). The release of phosphorylated NELF exposes a binding site for Polymerase Associated Factor 1 (PAF1), which converts Pol II to an elongation complex[29,30]. After pause-release, elongation factors and histone chaperones assemble on Pol II, and topoisomerase gains its full activity, preparing the transcription complex to progress through the array of nucleosomes and other obstacles including intragenic enhancers along the gene[31–34].

To track molecular modifications of engaged Pol II complexes at high specificity and nucleotide-resolution, we develop and describe Precision Run-On coupled to Immunoprecipitation sequencing (PRO-IP-seq). The PRO-seq portion of the protocol[15] labels active sites of nascent transcripts with biotinylated nucleotides, which allows affinity purification of nascent transcripts away from the non-nascent RNA that covers the chromatin, and identification of the precise genomic positions and orientations of engaged transcription complexes. The IP portion of the PRO-IP-seq protocol[35,36] selects Pol II complexes based on molecular modifications or composition using native immunoprecipitation. Here, we describe the biochemical principle of the PRO-IP-seq methodology and demonstrate how it tracks regulatory changes of the engaged transcription machinery at nucleotide-resolution. Taken together: The Pol II CTD remains unphosphorylated until the early promoter-proximal pause-region. Phosphorylation on serines 2 and 5 of the CTD occur preferentially at the late pause-region, between +31 nt to +60 nt from the initiation. Serine-7-phosphorylation of Pol II CTD dominates from the pause-release through to the +5 nucleosome, a region where Pol II assembles elongation factors and gains speed (reviewed in ref. 37). Our results confirm serine-2-phosphorylation as the most distinctive CTD mark of active elongation; however, serine-5-phosphorylation is also prevalent throughout the gene's body and correlated with higher transcriptional activity of a gene. Finally, tracking changes in Pol II CTD upon rapid transcriptional reprogramming by heat shock reveal that the CTD of paused Pol II can be, and often is, phosphorylated at serines 2 and 5.

## Results

### PRO-IP-seq tracks molecular modifications of nascent transcription complexes at nucleotide-resolution

Run-on reactions map engaged Pol II complexes at nucleotide-resolution by incorporating a single biotin-tagged nucleotide into the active sites of transcription[15]. To couple run-on reaction to immunoselection of Pol II populations, we performed the run-on reaction on chromatin[38]. Chromatin was prepared from three biological replicates of non-stressed (NHS) and 30 min at 42 °C heat shocked (HS30) human K562 cells and fragmented to <80 base pairs (bp) using light sonication and DNase I (Fig. 1a and Supplementary Fig. 1A). Since elongating Pol II footprints 43 bp of DNA in vitro[39], each fragment is expected to contain at most a single engaged Pol II complex. Even at promoter-proximal pause sites where Pol II can be at very high density, a single engaged Pol II sterically hinders new initiation

until transcription has proceeded beyond the pause[20,40]. Note also that DNase I does not cleave RNA and has a low activity towards RNA:DNA hybrids[41,42], leaving the nascent transcripts intact. Thus, the assay is designed to map individual Pol II complexes at the resolution and sensitivity afforded by PRO-seq. To obtain an external normalization control that follows through the protocol, we spiked-in mouse chromatin to each sample before the run-on reaction. This ensures that throughout the protocol, human and mouse chromatin are treated identically, including the biotinylation of 3'-ends of nascent RNAs during run-on reaction, immunoprecipitation of transcription complexes using antibodies that recognize both human and mouse Pol II CTD, isolation of biotinylated RNAs with streptavidin, and the subsequent library preparation of the nascent RNAs.

The run-on reaction was conducted in the presence of biotinylated A, C, G and U to obtain single nucleotide-resolution maps of engaged Pol II complexes, and the run-on chromatin was divided into distinct tubes for pull-downs with monoclonal antibodies (Fig. 1a and Supplementary Fig. 1B). The antibodies were previously verified to be specific against the distinct Pol II CTD modifications[43]. As a negative control, we used non-specific antibody (IgG), and as a positive control, we examined total nascent (PRO-seq) RNAs (noAb). In this strategy, the same chromatin and labeled nascent RNAs served as starting material for all antibody pulldowns and controls in a condition. After the immunoprecipitation, the nascent transcripts were purified and barcoded, and samples re-pooled to ensure identical handling (Fig. 1a). The size-distribution of reverse-transcribed nascent RNAs (Supplementary Fig. 1B) was as expected for nascent RNA-seq libraries[44], and the Pol II CTD pulldowns generated were deeply sequenced (Supplementary Fig. 1C), yielding nucleotide-resolution density profiles (Supplementary Fig. 1D) characteristic of nascent transcription. The IgG pulldown showed minute signal as total uniquely mapped reads (Supplementary Fig. 1C) and as density profiles (Supplementary Fig. 1D).

The data was initially normalized and examined with Reads Per Million of mapped reads (RPM) strategy (Supplementary Fig. 2A). To refine the normalization strategy, we created a normalization factor corrected control RPM (nf-cRPM) that does not assume a similar total transcription between the samples (see online Materials and Methods). The nf-cRPM yielded highly similar density profiles between PRO-IP-seq replicates, recapitulated known patterns of gene and enhancer transcription, brought distinct run-on experiments to comparable y-scale, and generated equivalent PRO-IP-seq profiles with different antibodies against the same Pol II CTD modification (Supplementary Fig. 1B, Supplementary Fig. 2B–D). After ensuring high statistical correlation (Supplementary Fig. 3A) and highly similar replicate profiles (Supplementary Fig. 3B), we merged the replicates into a single sample profile (Fig. 1b).

### PRO-IP-seq identifies + 1 nucleotides from nascent transcripts

Tracking nucleotide-resolution changes in transcription from initiation through the promoter-proximal pause requires precise identification of the +1 nucleotide (nt) and the active sites of transcription. The Transcription Start Sites (TSSs) derived from the reference genomes generally report the longest form of each mRNA isoform, which does not necessarily coincide with the most prominent nucleotide that initiates the transcription. To precisely map the Transcription Start Nucleotide (TSN, +1 nt) at each gene, we used the 5'-ends of nascent RNAs that associate with Pol II in our PRO-IP-seq data (Fig. 1c, Supplementary Fig. 4A–D). The 5'-ends of PRO-IP-seq reads enriched virtually at the same +1 nt regardless of the CTD modification (Supplementary Fig. 4A–C), which supports the concept that the CTD phosphorylation does not affect the site of initiation, and also that sites of initiation do not affect CTD phosphorylation patterns. In agreement, the same +1 nt was detected in distinct PRO-seq datasets that do not perform an antibody pulldown (Supplementary Fig. 4A–C). Worth

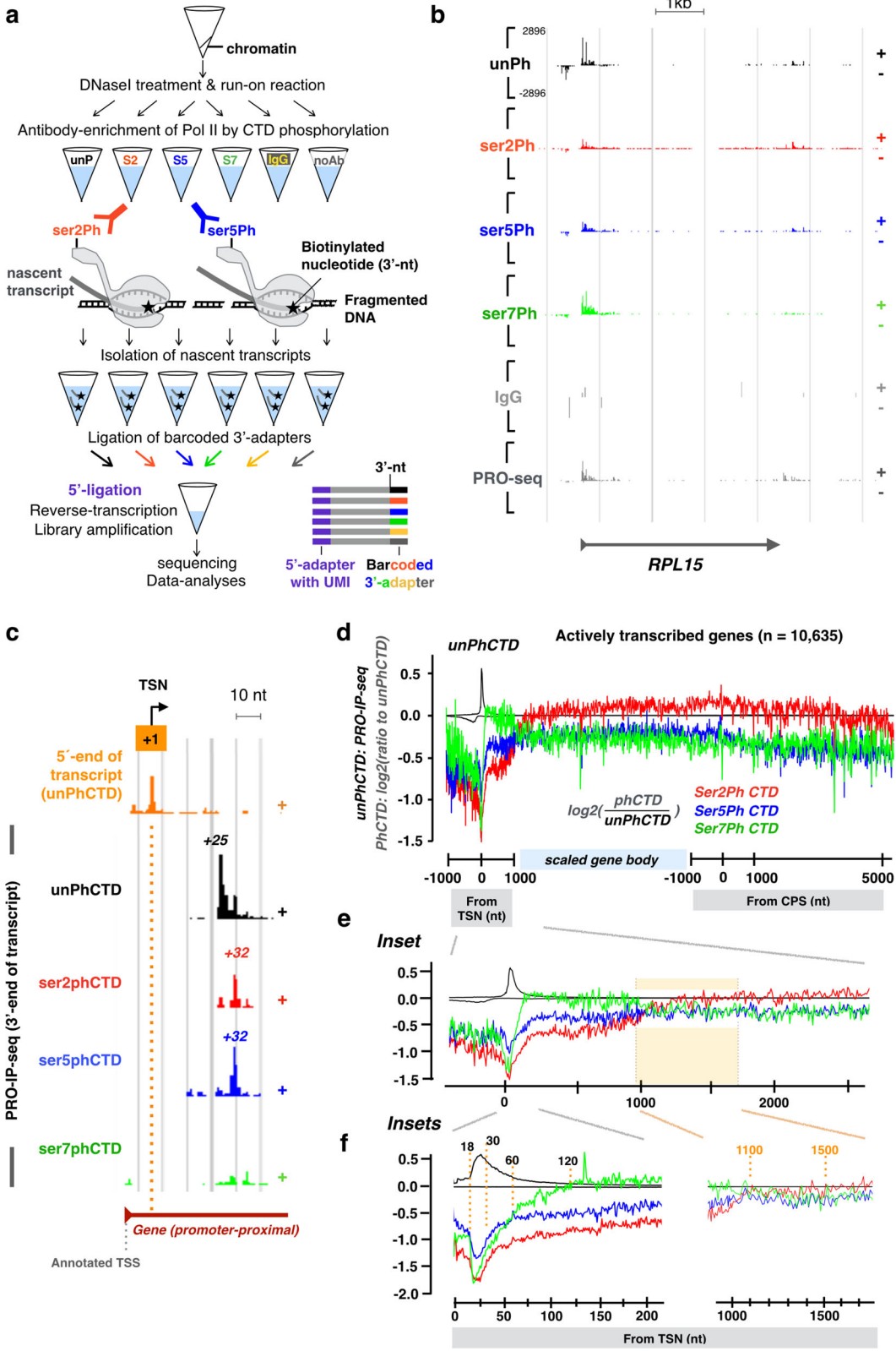

noting is that the TSN identified from 5′-ends of nascent transcripts resided on average 77 nt (median 55 nt) downstream of the reference genome annotated TSS (Supplementary Fig. 4B and D). The discrepancy between reference genome annotations and sequencing of 5′ ends of RNAs has been previously noted[16,45]. These and our current study highlight the importance of identifying the exact +1 nt from nascent transcripts in the cells being studied to provide a correct

reference point to map chromatin architecture and the precise genomic positions of the transcription machinery.

## Genes are divided into five domains of Pol II CTD modifications
Visualizing Pol II CTD modifications across individual genes uncovered high enrichment of Pol II with unphosphorylated CTD at the promoter-proximal pause region (Fig. 1b). Serine-2-phosphorylated CTD, instead,

**Fig. 1 | PRO-IP-seq maps molecular modifications of nascent transcription complexes at nucleotide resolution. a** Schematic presentation of the biochemical steps of PRO-IP-seq: Isolated chromatin is fragmented and a single biotinylated nucleotide incorporated into active sites of transcription. Transcription complexes are selected with antibodies against the C-terminal domain of Pol II (CTD), and nascent transcripts separated from non-nascent. The nascent transcripts are ligated to barcoded 3′-adapters, and the barcoded samples pooled for equal handling. After removal of 5′-cap, UMI-containing 5′-adapters are ligated. S2, S5 and S7 refer to phosphorylations of serine residues at Pol II CTD. **b** Representative example gene, *RPL15*, showing distribution of engaged Pol II with indicated status of CTD in non-treated K562 cells. The Y-scale is linear and the same in all tracks. **c** 5′-nt of PRO-IP-selected nascent RNAs enrich at the precise transcription start nucleotide (TSN; +1 nt), whereas the 3′-nt indicates the site of active transcription. The position of the

highest Pol II pause coordinate from the +1 nt is indicated for each Pol II CTD modification. **d** Comparison of distinct CTD phosphorylations relative to the unphosphorylated form of Pol II CTD across the promoter-proximal region (linear scale from −1000 to +1000 nt from the TSN, +1 nt), gene body (scaled to 50 bins per gene), and CPS & termination window (linear scale from −1000 to +5000 from the CPS). The CTD modifications were mapped across all actively transcribed genes in untreated human K562 cells ($n = 10,635$). Average PRO-IP-signal is indicated for unphosphorylated Pol II CTD (black transcription profile), and the ratio of phosphorylated relative to unphophorylated CTD (red, green and blue) shown in log2 scale. **e, f** Insets of the promoter-proximal and early gene body regions. The dotted orange lines indicate nt-resolution distance from the TSN. In (**e**–**f**), each window is 1 nt. The Y-scales are as in **d**.

was distributed over gene bodies, and serine-7-phosphorylated CTD occupied early coding regions (Fig. 1b). To broadly investigate changes in Pol II CTD phosphorylation across genes, we graphed the ratio of phosphorylated CTD against unphosphorylated CTD at all transcribed genes (Fig. 1d–f). These analyses uncovered five domains of Pol II CTD modifications (Fig. 1c–f): (1) At the early pause-region, the CTD was primarily unphosphorylated, while (2) in the late pause region, the CTD showed rapidly increasing phosphorylation on serines 2, 5, and 7. In the (3) immediate region downstream of the pause, the CTD contained strikingly high serine-7-phosphorylation, which was replaced by (4) high serine-2-phosphorylation along the region from 1.2 kb from initiation to the cleavage and polyadenylation site (CPS). Finally, (5) in the termination window, the CTD retained serine-2-phosphorylation but lost serine-5-phosphorylation. Changes in Pol II CTD occurred over short genomic distances, as demonstrated by the steep incline in serine-7-phosphorylation over promoter-proximal pause-release, and the prominent shift from serine-7- to serine-2-phosphorylation at +1000 to +1200 from the TSN (Fig. 1e, f).

The 52 heptad repeats in human Pol II can undergo multiple types of post-translational modifications (reviewed in[5]). Consequently, the availability of unphosphorylated CTD could be affected by modifications beyond serine-2, serine-5, and serine-7 phosphorylation. To ensure appropriate comparison of Pol II CTD state, we also analyzed the Pol II CTD modifications relative to the total level of transcription (PRO-seq only, noAb). These analyses showed highly similar patterns and densities of Pol II CTD modifications as detected when comparing phosphorylations relative to unphosphorylated CTD (Supplementary Fig. 5). Taken together, nucleotide-resolution PRO-IP-seq shows Pol II to initiate with unphosphorylated CTD, to become increasingly phosphorylated at serines 2, 5 and 7 as it progresses over the promoter-proximal pause region, to then be highly enriched for serine-7 phosphorylation from the point of pause-release to +1.2 kb from the initiation. After +1.2 kb from the initiation Pol II CTD becomes abundantly phosphorylated at serine-2 over the gene body and termination window, which is accompanied by reduced relative levels of serine-5-phosphorylation after the CPS (Fig. 1 and Supplementary Fig. 5).

### PRO-IP-seq is highly sensitive and specific for nascent RNAs
NET-seq[46] and its mammalian adaptation[47] have enabled seminal findings in Pol II CTD phosphorylation and transcriptional regulation (reviewed in refs. 5,6). Overall, PRO-IP-seq identified modifications at engaged Pol II (Fig. 1b–d) correlated well with previously reported Pol II phosphorylations, deduced from mNET-seq and ChIP-seq data (reviewed in refs. 5,6). However, mNET-seq selects Pol II associated RNAs, not all of which are nascent, giving rise to various signal peaks that are not present in run-on based assays (reviewed in[1]). Accordingly, PRO-IP-seq did not contain spurious spikes, which are present in various mNET-seq datasets, with or without the use of empigen (Fig. 2 and Supplementary Figs. 6–7). As an example, PRO-IP-seq did not enrich for engaged Pol II bearing serine-5-

phosphorylation at exon-intron boundaries (Fig. 2a–c, Supplementary Fig. 6F and Supplementary Fig. 7A), initially reported in mNET-seq[47] and later identified to be non-nascent RNA from the spliceosome[48]. Of note is that the spurious spikes in mNET-seq data extended from gene bodies (Fig. 2d–e, Supplementary Fig. 6A, B, Supplementary Fig. 7C) to promoter-proximal regions (Fig. 2f and h, Supplementary Fig. 6C and Supplementary Fig. 7D), obscuring the detection of Pol II initiation and pausing. In PRO-IP-seq, the double enrichment for a modification of interest and nascent RNA allowed identification of the precise initiation coordinate and Pol II pause sites. This is demonstrated by the single Pol II pause-region (Fig. 2a, c, and g, Supplementary Fig. 6F, G and Supplementary 7A, B) and precise +1 nts (Supplementary Fig. 4 and Supplementary Fig. 6G) in PRO-IP-seq data, compared to the several potential pause-regions (Fig. 2d, and f, Supplementary Fig. 6A–C and Supplementary Fig. 7C, D) and initiating nucleotides (Supplementary Fig. 6D, E) in mNET-seq assays. We conclude that nucleotide-resolution identification of the initiation and active sites of transcription from affinity-purified nascent RNAs enables tracking Pol II CTD modifications at unprecedented resolution and sensitivity (Figs. 1–2).

### Pol II CTD is primarily unphosphorylated until +30nt from the transcription start nucleotide
Early targeted-gene studies showed that genes contain two distinct promoter-proximal pause peaks within a single pause-region[13]. Later, coordinated, genome-wide mapping of initiation and pausing revealed two distinct enrichments of engaged Pol II at the promoter-proximal region: an early pause until +30 nt, and a late pause between +31 to +60 nt, from the TSN[16]. Here, we utilized the nucleotide-resolution tracking of CTD modifications and uncovered that unphosphorylated CTD dominates from the initiation until the early promoter-proximal pause-region (Figs. 1f and 3a, Supplementary Fig. 5 and Supplementary Fig. 7a, b). Indeed, unphosphorylated Pol II was most enriched at +25 nt from the TSN, after which, the relative levels of serine-2-, serine-5- and serine-7-phosphorylations increased (Fig. 1f and Supplementary Fig. 5). In accordance, Pol II with unphosphorylated CTD often occupied the pause closer to the TSN, while Pol II with serine-2- and serine-5-phosphorylated CTD was enriched at pause sites more downstream (Fig. 3a and Supplementary Fig. 7b). At highly transcribed genes, such as *beta-actin* (*ACTB*), unphosphorylated CTD was the primary form of paused Pol II detected across the pause region (Fig. 3b). Moreover, a clear pause signal was most often found with an antibody to the unphosphorylated CTD epitope (77% of all active genes), whereas pausing of Pol II bearing phosphorylated CTD was detected at 59% (serine-2 and serine-5), or 56% (serine-7) of transcribed genes (Fig. 3c). Across genes, unphosphorylated CTD enriched primarily at the early pause-region, peaking between +18 to +30 nt from the TSN (Fig. 3c), while Pol II with serine-2-, serine-5- or serine-7-phosphorylated CTD were more evenly distributed over the pause regions, showing a preference toward the late pause (Fig. 3c).

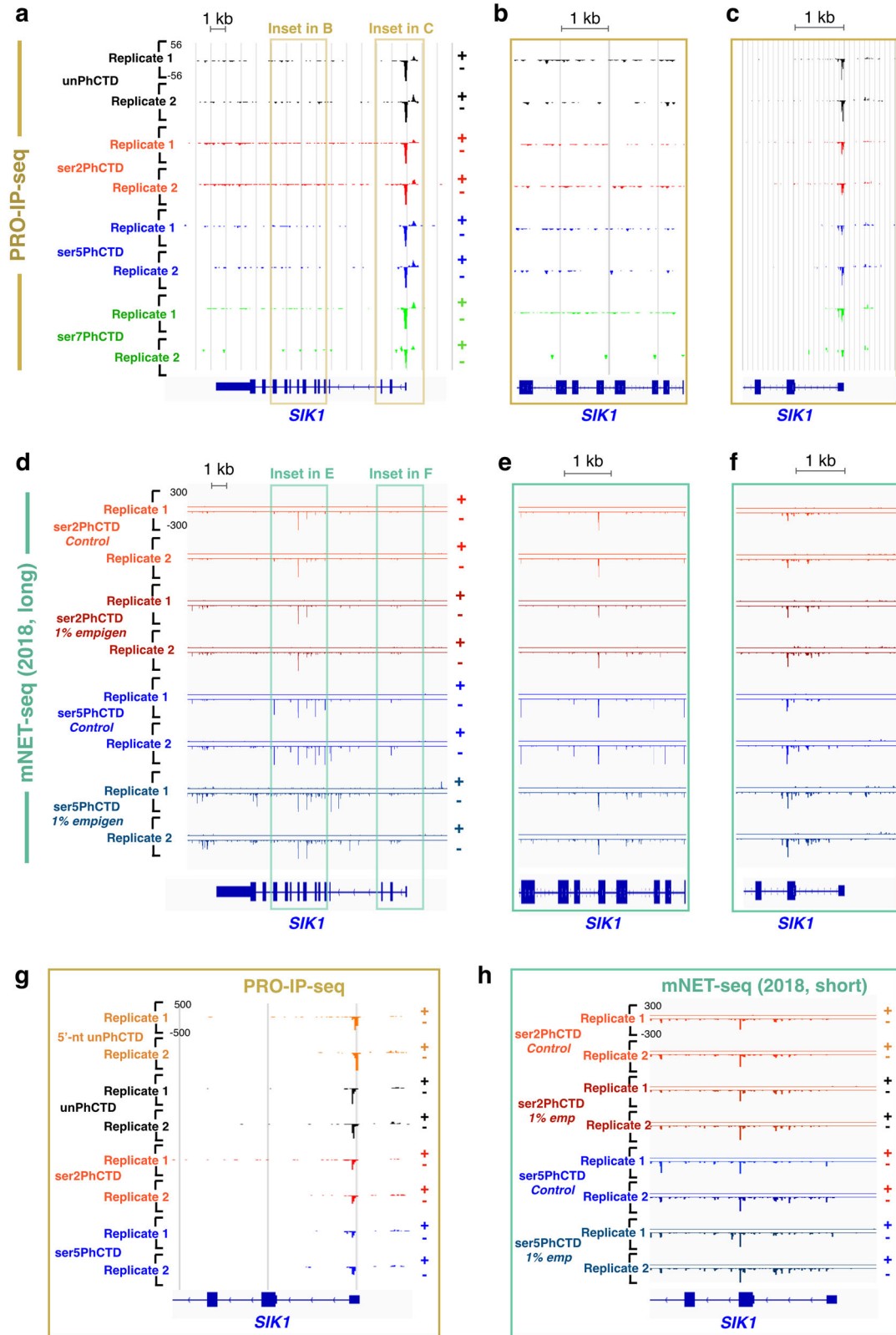

**Fig. 2 | PRO-IP-seq is highly selective for nascent transcripts. a–c** PRO-IP-seq and (**d–f**) mNET-seq profiles of transcription at *SIK1* gene. **b** and **e** zoom into the gene body, **c** and **f** into the promoter-proximal region, comparing PRO-IP-seq and mNET-seq at the same genomic regions. **g** 5'-nt (orange) and 3'-nt of PRO-IP-seq reads at

*SIK1* promoter. **h** 3'-nt of short mNET-seq fragments at *SIK1* promoter. The mNET-seq data was obtained as normalized bigWig files from Nojima et al. 2018 (GSE106881). The *SIK1* gene has previously been shown as an example gene for expected mNET-seq signal[89].

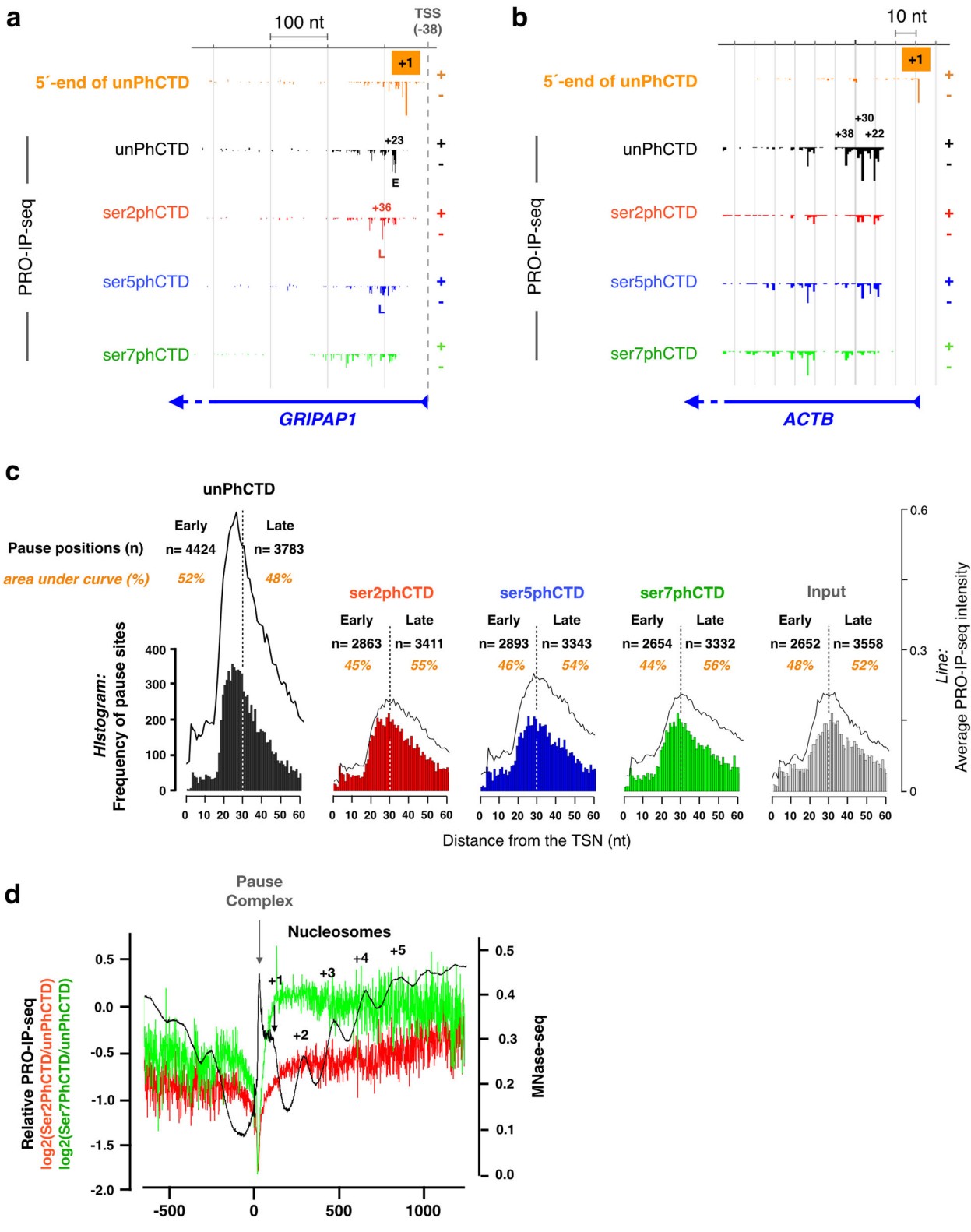

## Serine-7-phosphorylation of Pol II CTD dominates from the pause-release until +5 nucleosome

Pol II CTD had low levels of serine-7-phosphorylation at the early pause-region (Fig. 1f). Moreover, serine-7-phosphorylated Pol II did not show prominent enrichment at either early or late pause site (Fig. 3a, b and Supplementary Fig. 7b). Instead, phosphorylation at serine-7 rapidly increased after the promoter-proximal pause, becoming the most prominent CTD phosphorylation after +60 nt from the TSN, and reaching its highest levels in +140 to +160 nt window (Fig. 1f). In this early coding region, Pol II assembles elongation factors, gains speed, and encounters the +1 nucleosome (reviewed in ref. 37). To investigate Pol II CTD phosphorylation with respect to the nucleosomes, we overlayed MNase-seq to the relative enrichments of serine-2- and serine-7-phosphorylations (Fig. 3d), centering the signal to the TSN.

**Fig. 3 | Pol II with unphosphorylated CTD is enriched at the early pause.**
**a, b** PRO-IP-seq identifies transcription start nucleotide (TSN, +1) and the exact pause sites of engaged Pol II complexes, as exemplified at *GRIPAP1* (**a**) and *ACTB* (**b**) promoters. At *GRIPAP1* Pol II with unphosphorylated CTD is enriched at early pause (E), while Pol II with serine-2-posphorylated CTD localizes to the late (L) pause. At *ACTB* unphosphorylated CTD has the strongest enrichment on all the pause sites. The TSN is identified using 5'-ends of PRO-IP-seq reads from the unPhCTD.The numbers indicate genomic distance (nt) from the TSN. The RefSeq-annotated transcription start site (TSS, dashed gray line) is depicted at *GRIPAP1* and its

distance to the TSN indicated. **c** Histograms showing the distance of the strongest pause coordinate to the TSN. The count (n) of pause sites detected at early (+1 to +30 from the TSN) and late (+31 to +60 from TSN) pause-region are indicated. The line above the histogram shows Pol II density across the pause-region. The percentage of Pol II at the early and late pause are counted as areas under the curve. **d** Relative levels of serine-2- and serine-7-phosphorylated CTD overlaid on MNase-seq detected nucleosomes at the open chromatin region of gene promoters. The PRO-IP-seq and MNase-seq were quantified on all transcribed genes in human K562 cells (*n* = 10,635). The MNase-seq data is from ENCODE Consortium[87].

These analyses uncovered serine-7-phosphorylation to peak at the +1 nucleosome and remain high until +5 nucleosome at the end of the exonuclease-accessible region (Fig. 3d). Serine-2-phosphorylation, instead, increased through the ordered array of +1 to +5 nucleosomes becoming the dominant Pol II CTD modification at the end of the MNase-accessible region (Fig. 1f and Fig. 3d).

### Actively transcribed gene bodies have high levels of serine-2 and serine-5-phosphorylated CTD

At an average gene, serine-2-phosphorylation was the most dominant CTD modification from +1200 nt from the TSN to the end of the termination window (Fig. 1b–f). However, many highly transcribed genes had prominent levels of serine-5-phophorylation across the gene body (Fig. 4a, b). To investigate whether serine-5-phosphorylation co-occurred with high gene activity, we grouped genes based on productive elongation (Fig. 4c, Supplementary Fig. 8A, B). While genes with low and moderate activity were covered by serine-2-phosphorylated CTD along the gene body (Fig. 4b, c), genes with the highest transcriptional activity contained comparable levels of serine-2- and serine-5-phosphorylation at the CTD (Fig. 4a, c). In contrast, the CPS and termination window were dominated by serine-2-phosophorylated CTD in every gene group studied (Fig. 4a–c).

### Serine-7-phosphorylation at the early coding region predicts productive elongation along the gene

Analyzing Pol II CTD modifications with respect to transcriptional activity uncovered serine-7-phosphorylation at the early coding region to predict the level of productive elongation downstream at the gene body (Fig. 4c and Supplementary Fig. 8B). The more productive elongation the gene had, the earlier the phosphorylation of serine-7 occurred and the higher its relative enrichment became (Fig. 4c). As comparison, relative levels of serine-2 and serine-5-phosphorylations were similar across the early coding regions regardless of the gene's transcriptional activity (Fig. 4c). Next, we overlaid MNase-seq and histone 4 acetylation (H4ac) ChIP-seq with the PRO-IP-seq signals of distinct gene groups. The positioning of +1 to +5 nucleosomes was comparable regardless of the transcriptional activity of the gene (Supplementary Fig. 8C), but their active histone marks, indicated as H4ac levels[49–51], increased with the productive elongation through the gene (Supplementary Fig. 8D). Increased chromatin activity at serine-7-rich regions was further supported by the positive correlation of Chromodomain Helicase DNA binding protein 1 (CHD1), and negative correlation of H2A.Z (Fig. 4d). Of note is that the accessibility of the promoter and the early coding region to DNase I was similar regardless of the gene's transcriptional activity (Fig. 4d). These results suggest that serine-7-phosphorylation at the CTD is involved in establishing the processivity of Pol II through the gene. This processivity is gained at the early coding region where elongation factors assemble on Pol II and the transcription elongation rapidly accelerates[19]. Indeed, the study of Pol II elongation rates by Jonkers *et al.* showed that Pol II undergoes its major acceleration during the first 1–2 kb. In accordance, once Pol II had proceeded beyond +1500 nt from the TSN, the levels of serine-phosphorylations did not change appreciably along the gene, either when analyzed as average enrichments over unphosphorylated

CTD or from individual genes (Fig. 1b and 1d, Fig. 4a–c, Supplementary Fig. 1D, Supplementary Fig. 2, Supplementary Fig. 3B and Supplementary Fig. 7A).

### Heat shock reorganizes Pol II CTD phosphorylation across the genome

Heat shock triggers an instant reprogramming of transcription[23–25,52–54], primarily coordinated at the rate-limiting step of Pol II pause-release (reviewed in[54]). At thousands of heat-repressed genes, Pol II accumulates at the promoter-proximal region, while heat-activated genes overcome the global repression by recruiting potent *trans*-activators that trigger the release of paused Pol II into elongation. One of these potent *trans*-activators is Heat Shock Factor 1 (HSF1) that can release paused Pol II at chaperone, co-chaperone, and polyubiquitin genes[55–57] (reviewed in ref. 58) via binding to proximal and distal regulatory elements[26,53,59]. To address how heat shock changes Pol II CTD modifications genome-wide, we utilized PRO-IP-seq to track Pol II CTD phosphorylations at (1) heat-induced genes where paused Pol II is efficiently released into elongation, and (2) heat-repressed genes where the release of paused Pol II is inhibited.

Heat shock induced profound changes in the phosphorylation of Pol II CTD at individual genes (Supplementary Fig. 9 and Supplementary Fig. 10A, B) and across topologically associated domains (TADs; Fig. 5 and Supplementary Fig. 10C). Intriguingly, primary TADs with highly heat-induced genes recruited more Pol II that came available from downregulated genes within the same TAD (Supplementary Fig. 10D). As a potential compensatory mechanism, TADs with major heat-induced loci contacted distant primary TADs with clusters of heat-repressed genes, as exemplified with *HSP70* and *HSP90* loci, both contacting a strongly heat-repressed histone locus (Fig. 5a–d). Indeed, histone genes in all three major clusters showed Hi-C and ChIA-PET identified connections to *HSP70* and *HSP90* genes (Fig. 5b, c). At the histone loci, all Pol II CTD modifications were reduced upon stress, and the transcription machinery concentrated at the promoter-proximal pause (Fig. 5d and Supplementary Fig. 9B). Concordantly, the HSP loci gained a massive transcriptional activation involving all Pol II CTD modifications (Fig. 5d and Supplementary Fig. 9A). The kinetic link between *HSP* induction and *histone* repression has been previously shown in *Drosophila*[60], mouse[24] and human[25] cells. Taken together, genes with similar responses can be separated into their own sub-TADs (Supplementary Fig. 10D), but we speculate that connections between TADs of opposite regulation may facilitate the diffusion of Pol II and its regulators from heat-repressed to heat-induced genes to help mediate extremely rapid shifts in transcription.

### Heat shock causes a global increase in serine-5-phosphorylated CTD

Heat shock is known to increase the proportion of phosphorylated, high-molecular weight, Pol II[61,62]. We found that the total cellular pool of Pol II gained a modest but statistically significant increase in phosphorylation at serine-5 of the CTD during heat stress (Fig. 5e, f and Supplementary Fig. 11A). Also, serine-2-phosphorylation increased during heat shock, but the increase was mainly detected upon 60 min

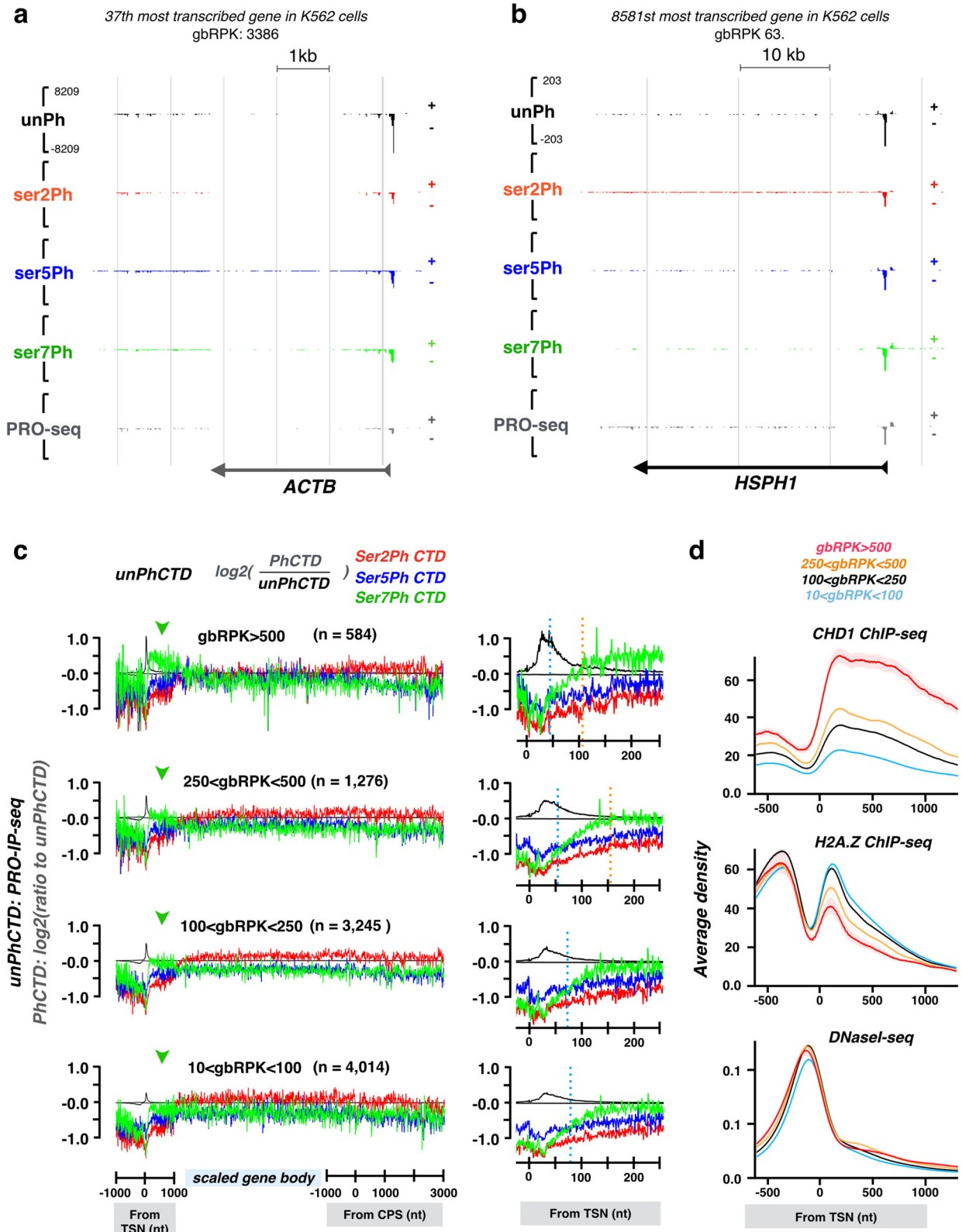

of heat stress (Fig. 5e, f and Supplementary Fig. 11A). Tracking phosphorylation of Pol II across the genome showed heat-induced genes to contain high levels of both serine-2- and serine-5-phosphorylated CTD (Supplementary Fig. 6A and Supplementary Fig. 11B). Even at heat-repressed genes, the Pol II that did enter productive elongation was enriched with serine-5 phosphorylation of the CTD (Supplementary Fig. 10B and Supplementary Fig. 11C).

## Pol II CTD gains serine 2 and serine 5 phosphorylation at the pause-region of heat-induced genes

Heat shock increases recruitment of Pol II to heat-activated genes[24,25]. Interestingly, the promoter-proximal regions of heat-activated genes gained primarily Pol II bearing serine-2- and serine-5-phosphorylated CTD (Fig. 6a). Moreover, the unphosphorylated Pol II shifted from the early to the late pause (Fig. 6b, c). In comparison, serine-7-

**Fig. 4 | Transcriptional activity is reflected in the CTD code. a** *Beta-actin (ACTB)*, one of the most active genes in human, shows (1) prominent pausing of Pol II, primarily with unphophorylated CTD, (2) ser-7-phosphorylation extending to early coding region, and (3) prominent ser-5-phosphorylation of the CTD across the gene body. **b** *HSPH1*, a lowly transcribed gene in non-stressed human cells, shows (1) high Pol II pausing that is detected with all the investigated antibodies, (2) no enrichment of ser-7-phosphorylation at the early coding region, and (3) ser-2-phosphorylation dominating across the gene body. **c** Genes were grouped based on their transcriptional activity (reads per kilo bases of gene body, gbRPK) and the CTD phosphorylations compared against the unphospharylated form of the CTD. Promoter-proximal region: linear scale from −1000 to +1000 nt from transcription start nucleotide (TSN). Gene body: scaled to 50 bins per gene. CPS & termination window: linear scale from −1000 to +5000 from the CPS. Average density of unphosphorylated Pol II CTD is indicated as black transcription profile, and the ratio of phosphorylated per unphophorylated CTD (red, green and blue) shown in log2 scale. The insets show the early coding region. The dotted blue vertical line indicates where the relative amount of ser-7-phosphorylation exceeds the relative amount of ser-5-phosphorylation. The dotted orange vertical line indicates where the relative amount of ser-7-phosphorylation exceed the relative amount of unphosphorylated CTD. **d** CHD1 ChIP-seq, H2A.Z ChIP-seq and DNaseI-seq signal[87] at the promoter-proximal region of indicated genes. The shaded area indicates 12.5–87.5% interval for each group, centered to the average density.

phosphorylation only modestly increased along the gene, and Pol II with serine-7-phosphorylated CTD did not appear to pause at the promoter-proximal region (Fig. 6a, b). These results support our nucleotide-resolution profiles in the NHS condition (Figs. 1–4), indicating that phosphorylation of serine 7 at the CTD occurs after Pol II has been released from the promoter-proximal pause into early coding region.

### Pol II with phosphorylated CTD remains paused at heat-repressed genes

Next, we investigated how the phosphorylation status of CTD changes at repressed genes where heat shock prevents the entry of Pol II into productive elongation. Strikingly, inhibiting the pause-release caused accumulation of Pol II with serine-2 and serine-5 phosphorylated CTD at the early and late pause regions of heat-repressed genes (Fig. 7a, b). The pausing of Pol II with phosphorylated CTD was particularly evident at long genes (Fig. 7c); Pol II molecules that were downstream of the pause when the heat stress began continued transcribing, causing a wave of clearing Pol II molecules from the coding regions (Fig. 7c). These clearing Pol II molecules maintained their phosphorylation status. In contrast, levels of Pol II at the pause-region increased during heat shock and the pattern of CTD phosphorylation changed. Pol II with unphosphorylated CTD epitope was enriched in the early pause in NHS condition, whereas Pol II with serine-2 and serine-5-phosphorylated CTD epitope occupied the late pause (Fig. 7c). Upon heat shock, the pause-release was inhibited and Pol II bearing unphosphorylated, serine-2-phosphorylated and serine-5-phosphorylated CTD concentrated to the early and late pauses (Fig. 7c). These results demonstrate that Pol II with serine-2- and serine-5 phosphorylated CTD can remain paused.

Despite the massive accumulation of Pol II to promoter-proximal pause at heat-repressed genes, serine-7 phosphorylation at the pause remained minimal (Fig. 7a–c). This indicates that even heat-triggered pausing of Pol II at the promoter-proximal region does not stabilize Pol II bearing serine-7-phosphorylation. Thus, serine-7 phosphorylation appears to be a modification intimately connected to the Pol II pause release, and the following assembly of elongation complex that prepares Pol II to its journey through the gene.

## Discussion
### PRO-IP-seq provides a nucleotide-resolution tool to discover dynamic changes in engaged Pol II
Pol II is a multi-subunit protein complex whose composition and molecular modifications change as RNA synthesis proceeds through the rate-limiting steps of transcription and transcription coupled RNA processing. Pol II progression at genes and enhancers has been analyzed at nucleotide-resolution (reviewed in ref. 1), and structures of the PIC, as well as pause and elongation complexes are available in elegant detail in vitro (reviewed in ref. 63). We have lacked efficient tools to monitor nucleotide resolution changes in Pol II complexes in vivo. Multiple techniques use immuno-selection of chromatin-bound Pol II, some of which sequence the associated RNAs to track RNA synthesis (reviewed in ref. 1). These techniques include Pol II ChIP-seq[64], NET-

seq[46] and mNET-seq[47] that have uncovered serine-2-phosphorylation of Pol II CTD to enrich at the 3′-ends of genes while serine-5- and serine-7-phosphorylations dominate at genes' 5′-regions (reviewed in ref. 5). None of the previous techniques combine nascent transcript labeling and selection of transcription machineries in distinct molecular states to achieve the sensitivity to map the nucleotide-resolution changes in Pol II CTD code. The PRO-IP-seq technique described here (1) labels the active sites of transcription, (2) generates chromatin fragments with a single engaged transcription complex, (3) immuno-selects Pol II by its modifications, (4) purifies the nascent transcripts, (5) maps the active sites of transcription and (6) the precise +1 nts from nascent transcripts, and (7) uses normalization strategy that does not constrain samples to the same total transcript count (Fig. 1 and Supplementary Figs. 1–2). As a result, the precise positional changes in Pol II modifications are mapped from initiation to early and late promoter-proximal pauses and into elongation and transcription termination (Figs. 1 and 3 and Fig. 4), and reveal a positional control of Pol II CTD modifications at high resolution and sensitivity (Fig. 8). We expect PRO-IP-seq to be adaptable to also investigate the molecular composition of transcription machineries across the genome, including the dynamic association and disassociation of pausing and elongation factors.

### Nucleotide-resolution CTD code from initiation to promoter escape
In vitro studies indicated that Pol II initiates transcription with unphosphorylated CTD[7]. After initiation, Cdk7 phosphorylates serine-5 of the CTD enabling Pol II to break ties with the PIC and the Mediator[10,11]. Our genome-wide analyses globally corroborate the Pol II that reaches the early pause shortly after initiation has an unphosphorylated CTD (Figs. 1 and 3 and Fig. 4). Furthermore, Pol II with unphosphorylated CTD peaks at +25 nt and remains high until +30 nt from the initiation. This Pol II at the early pause appears associated with the PIC and has not gone beyond the point of TFIID interaction[15,65,66]. At the promoter-proximal region, serine-5 phosphorylation enhances 7-methylguanylate capping of the nascent transcript[67–69]. The early pause-region has been associated with low, and the late pause-region with high levels of 5′-capping[13,16], indicating high-resolution co-occurrence of serine-5-phosphorylation and 5′-capping across the genome (Fig. 3).

### What functions as a molecular switch to release the promoter-proximally paused Pol II?
Serine-5, and to a lower extent serine-2, of the CTD was phosphorylated within the pause-region, primarily between +31 to +60 nt from the initiating nucleotide (Fig. 1F, Fig. 3a and c). Since Pol II with unphosphorylated CTD occupies the early pause region (+18 to +31), it's tempting to speculate that CTD phosphorylation increases the likelihood of Pol II to progress through the pause-region and release to productive elongation. However, provoking a transcriptionally-repressive state at several thousand genes using heat shock revealed that phosphorylation of CTD at serine-2 and serine-5 residues is not sufficient to launch Pol II into elongation (Fig. 7). Indeed, the Pol II that

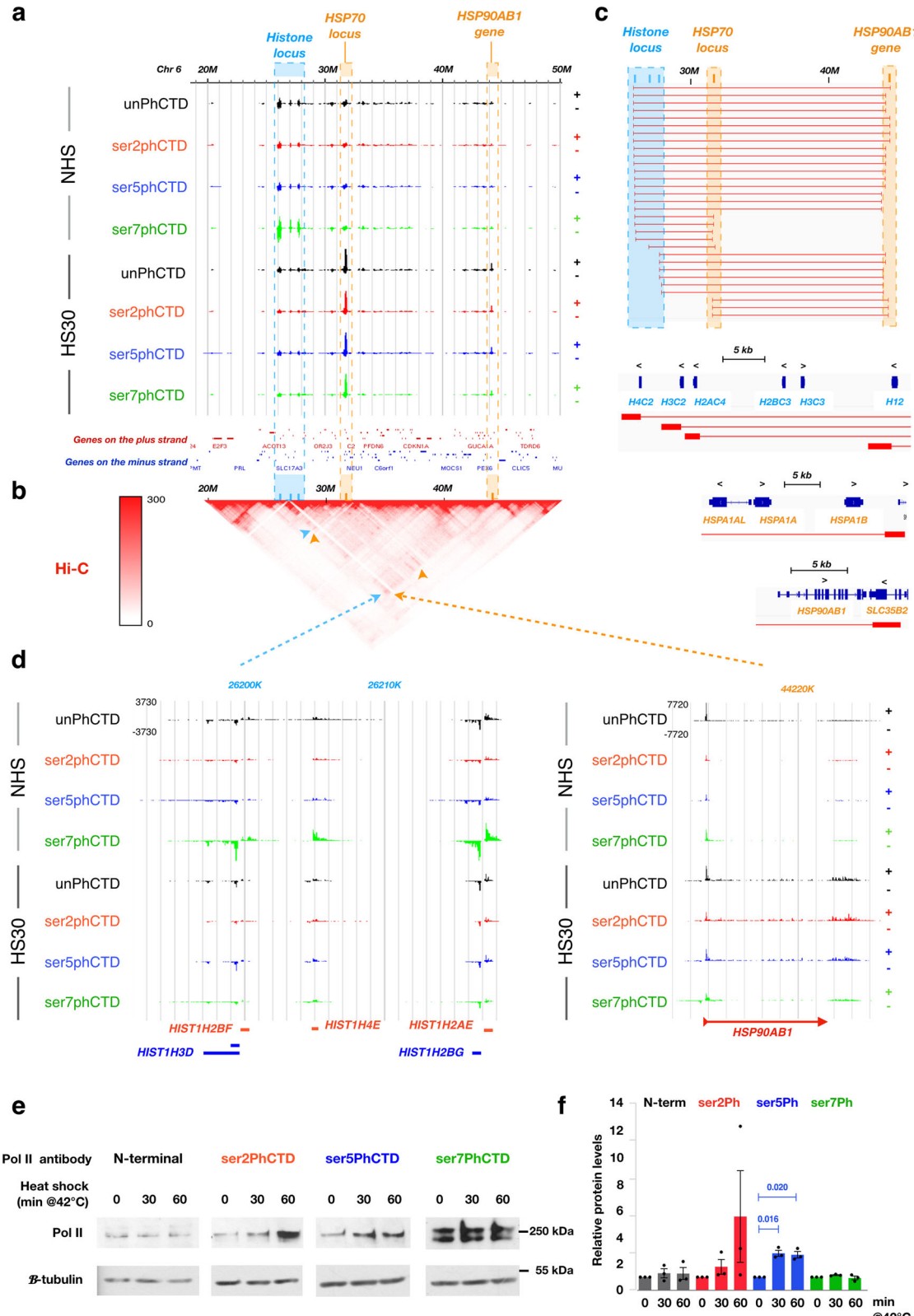

**Fig. 5 | Heat shock reprograms Pol II CTD phosphorylation. a** A 30-Mb region of chromosome 6, showing positions of *histone, HSP70 and HSP90 loci* and changes in Pol II CTD phosphorylation (**b**) Hi-C[84] map indicating connections between *HSP* and *histone loci* within a nested TAD. **c** ChIA-PET-identified[86] connections (red blocks and lines) that occur between *histone, HSPA1A-HSPA1B* and *HSP90AB1* genes. Expanded browser profiles below the connection data exemplify connections originating from the *histone, HSPA1A-HSPA1B,* and *HSP90AB1 loci*. **d** Part of the histone locus (left panel) and the *HSP90AB1* gene (right panel) showing changes in Pol II CTD. **e** Representative Western Blotting and (**f**) quantification of total Pol II and CTD phosphorylations from three Western Blot replicates. Standard error of mean and *p*-values < 0.05 (student's *t*-test) are shown. Source data for WB images and quantifications are provided as Source Data 1-3.

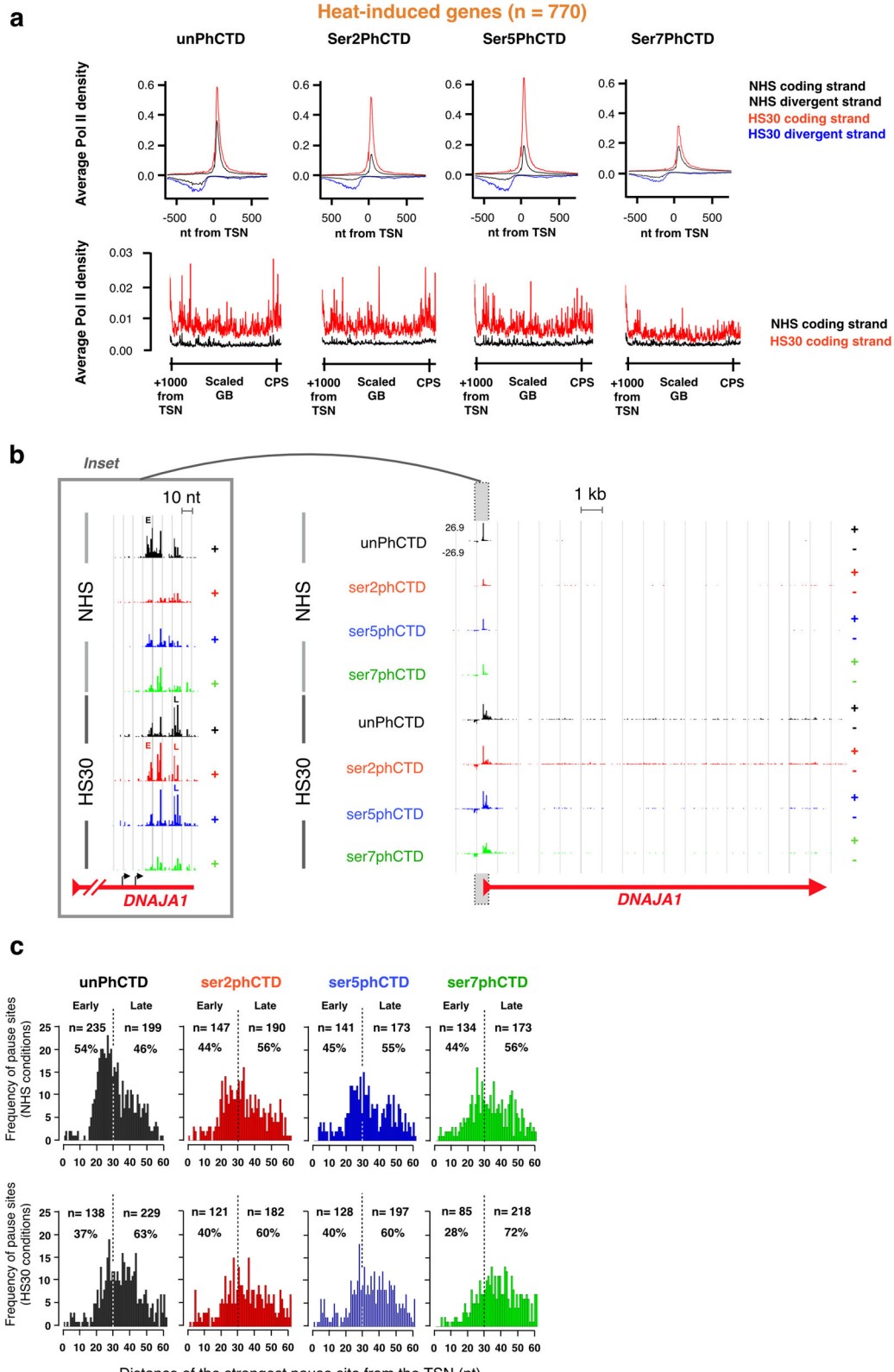

**Fig. 6 | Heat-activation launches Pol II into elongation and concentrates promoter-proximal Pol II to distal pause. a** Average PRO-IP-seq densities with indicated Pol II CTD modification at promoter-proximal regions (upper panels) and gene bodies (lower panels) of heat-activated genes (*n* = 770). The black lines depict PRO-IP-seq signal in non-stressed (NHS) and red (coding strand) and blue (non-coding strand) in 30 min heat shocked (HS30) cells. TSN Transcription start nucleotide. **b** Distribution of CTD modifications on *DNAJA1* gene showing heat-induced wave of Pol II with all detected CTD marks. The inset shows promoter-proximal region, depicting the distribution of CTD modifications to early (E) and late (L) pauses. The black arrows indicated two distinct +1 nucleotides (transcription start nucleotides, TSNs) detected from the nascent transcripts. **c** Histograms depicting the coordinate of the highest pause on each promoter-proximal region of heat-induced gene in NHS (upper panels) and upon HS30 (lower panels). The count (*n*) and fraction (%) of pauses detected on early ( + 1 to +30 from the TSN) and late (+31 to +60 from TSN) are indicated.

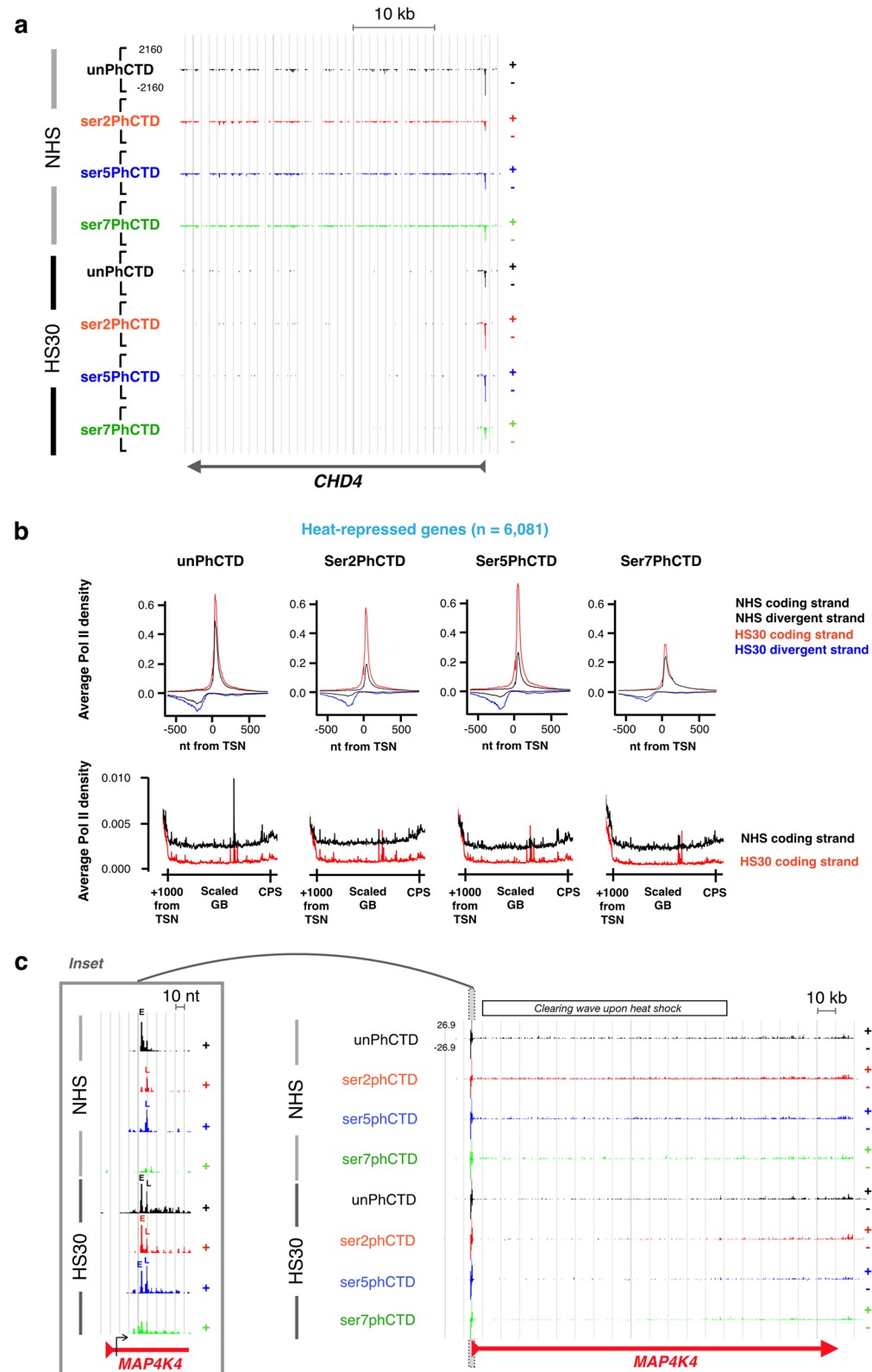

**Fig. 7 | Heat shock captures Pol II with phosphorylated CTD to proximal and distal pauses. a** Heat shock represses *CHD4* gene by capturing Pol II to the promoter-proximal region and clearing transcription complexes from the gene body. Please note that the heat shock-triggered increase in promoter-proximal pausing constitutes primarily of serine-2 and serine-5-phospohorylated CTD. **b** Average PRO-IP-seq densities with indicated Pol II CTD at promoter-proximal regions (upper panels) and gene bodies (lower panels) of heat-repressed genes. The black lines depict PRO-IP-seq signal in non-stressed (NHS) and red (coding) and

blue (non-coding) in 30 min heat shocked (HS30) cells. TSN: Transcription start nucleotide. **c** Distribution of CTD modifications on *MAP4K4*. Please note that *MAP4K4* is a long gene where the clearing of Pol II from the gene body has not reached the end of the gene during a 30-min heat shock. The Pol II clearing wave is indicated above the graph. The inset shows distribution of CTD modifications at early (E) and late (L) pauses. The black arrow shows the +1 nucleotide detected from the nascent transcripts.

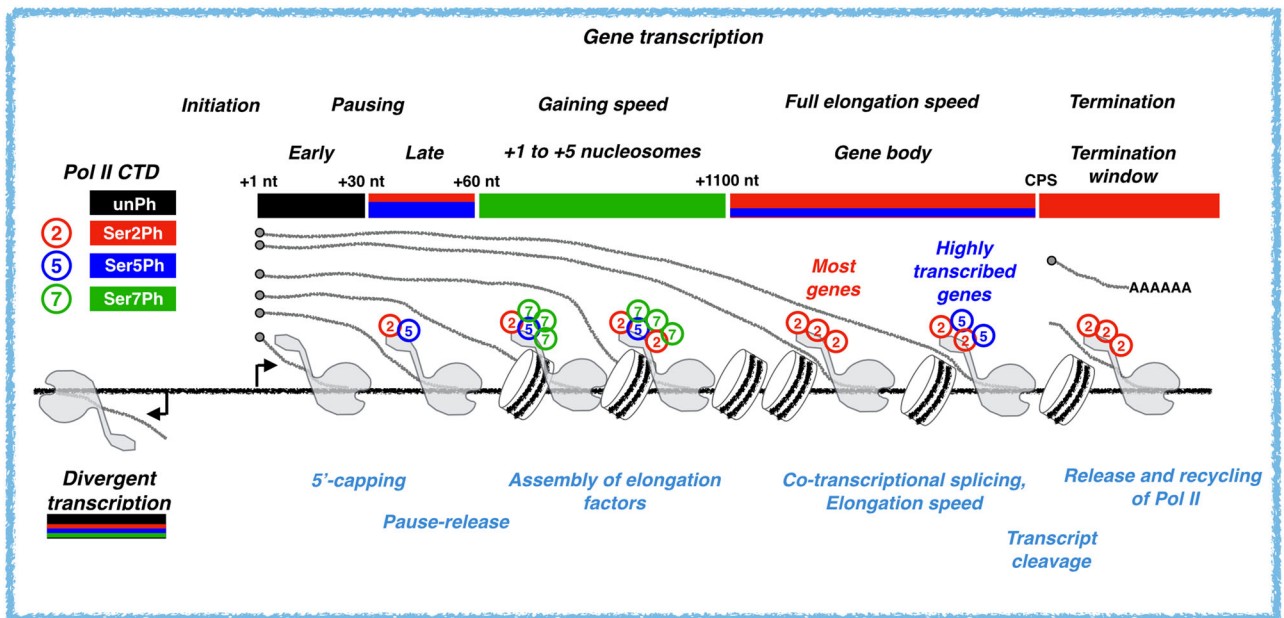

**Fig. 8 | Phosphorylation of Pol II CTD from initiation to transcription termination.** Schematic representation of Pol II CTD phosphorylation status as the transcription proceeds from the initiating nucleotide (+1 nt), through early and late promoter-proximal pause regions, into open chromatin across +1 to +5 nucleosomes, gene body and termination window. The major CTD modifications are indicated with color code above the gene, and co-transcriptional processes occurring at each region listed below the gene.

accumulated at the pause-regions of heat-repressed genes was abundantly phosphorylated at serines 2 and 5 (Fig. 7). These results highlight the need to track the detailed regulatory mechanisms that stabilize Pol II pausing and trigger its release into premature termination or productive elongation. In metazoan species, the release of the Pol II into elongation requires P-TEFb that phosphorylates NELF, SPT5 and serine-2 of Pol II CTD (reviewed in ref. 37). Particularly, the CDK9 mediated phosphorylation of SPT5 has been identified as the key trigger to release paused Pol II[70]. Besides functioning as a signaling platform, CTD phosphorylation has been suggested to aid Pol II escape from a phase-separated domain or condensate around the promoter[71]. NELF has been reported to form condensates when overexpressed in heat stressed cells, and reduced release of NELF coincides with heat-repression[72]. Coupling nucleotide-resolution changes in Pol II composition to the progression of RNA synthesis through the rate-limiting steps of transcription should be highly informative in clarifying the regulatory logic of Pol II pause-release, including the functional consequences of P-TEFb, as well as serine-2, NELF and SPT5 phosphorylation.

### Early coding region prepares Pol II to elongation
The median human gene is 20,000 nt long, requiring Pol II to proceed considerable genomic distances to transcribe a pre-mRNA. After being released from the promoter-proximal pause, Pol II proceeds slowly (0.5 kb/min), gaining nearly its full elongation speed (2–5 kb/min) after +1000 nt from the TSS (reviewed in ref. 37). We found serine-7-phosphorylation to rapidly increase around the pause-release, peaking at the +1 nucleosome, and remaining high until the end of the MNase-accessible region, +1000 nt from the initiation (Fig. 3c). The serine-7-phosphorylation positively correlated with histone acetylation, presence of chromatin modifiers, and intriguingly, predicted transcriptional activity and mRNA abundance (Fig. 4 and Supplementary Fig. 11B). Serine-7-phosphorylation has been suggested to prime CTD for recognition by P-TEFb, enabling serine-2-phosphorylation of CTD only at the appropriate stage of the transcription cycle[73]. These results suggest that distinct kinases could phosphorylate serine-2 of the CTD before and after the pause-release. Indeed, serine-2-phosphorylation

was found as a universal mark of active transcription along the gene bodies (Figs. 1c and 4a–c). However, serine-5-phosphorylation was enriched along actively transcribed genes in non-stressed cells (Fig. 4), and along genes that escaped the transcriptional repression upon heat shock (Supplementary Fig. S11C). Intriguingly, serine-5-phosphorylation has been coupled to co-transcriptional splicing[48], and transcription upon heat shock is known to increase the rate of Pol II elongation to an average 3.5–4.7 kb/min[24,74], suggesting that serine-5-phosphorylation might ensure a fast production of mature mRNA.

In summary, PRO-IP-seq provides a nucleotide-resolution tool to track the molecular modifications of engaged transcription machineries. It uncovered several components of the regulatory logic of Pol II CTD phosphorylation as RNA-synthesis proceeds through rate-limiting steps of transcription. Our study lays the framework to track how Pol II phosphorylation states interplay with its molecular composition and regulation at nucleotide-resolution across the genome.

### Limitations
PRO-IP-seq reports nascent transcripts that associate with transcription machineries in distinct molecular states. The enrichment of Pol II populations in this study is based on antibodies raised against epitopes of Pol II CTD. As with any antibody dependent technique, PRO-IP-seq relies on the specificity of the antibodies. The antibodies used in this study were raised against epitopes containing the same amino acid sequence. For example, the 8WG16 antibody was raised against unphosphorylated heptad repeats of Pol II CTD. Although it has been shown to have high specificity to the unphosphorylated Pol II CTD, it likely also detects small amounts of phosphorylated forms of Pol II CTD. This possibility of low level of cross-recognition applies to all the CTD targeting antibodies in this study. To obtain high reliability, we used antibodies that had previously been validated for epitope specificity. Furthermore, for both serine-2 and serine-5 phosphorylation two distinct antibodies were used. Another layer of complexity with Pol II CTD comes from the 52 repeats of $T_1S_2P_3T_4S_5P_6S_7$ present in the human Pol II CTD: Distinct repeats within a single CTD can bear different post-translational modifications. Our study can report the relative levels of each detectable Pol II modifications at nucleotide-

resolution across the genome, but it cannot resolve the combinatorial code arising from distinct heptad repeats bearing different serine modifications at a given CTD.

## Methods

### Cell culture and heat shock treatments

Human K562 erythroleukemia cells, originating from ATCC and obtained[57] from Prof. Lea Sistonen laboratory, Åbo Akademi University, Turku, Finland, were cultured in RPMI (Sigma). The spike-in wild-type MEFs[24] originated from Prof. Susan Lindquist laboratory, MIT, Boston, USA, and were cultured in DMEM (Gibco). RPMI and DMEM were supplemented with 10% fetal calf serum, 2 mM L-glutamate, 100 µg ml⁻¹ streptomycin, and 100 U ml⁻¹ penicillin. Both cell lines were maintained at 37 °C with 5% $CO_2$ and 90% humidity. Both cell lines were tested to be mycoplasma free, and to display morphology, proliferation rate and transcriptional profile characteristic to the respective cell line. The heat shocks in K562 cells were instantly provoked by resuspending 30 million pelleted cells in pre-conditioned pre-warmed media (42 °C) and maintaining the cells at 42 °C in a water bath for 30 min. 30 million non-heat shocked cells were resuspended in 37 °C pre-conditioned pre-warmed media and maintained at 37 °C for 30 min. To avoid provoking transcriptional changes by freshly added media, the cells were expanded 24 h prior to the treatments. The pre-conditioned media was collected from excess cells of the same culture prior to the experiments[75].

### Chromatin isolation

Cells were collected by transferring the cell suspension from 37 °C or 42 °C into eight times of the volume of ice-cold PBS. The cells were pelleted by centrifugation at 4 °C and washed with ice cold PBS. The cell pellets were resuspended in 1 mL ice-cold NUN buffer (0.3 M NaCl, 1 M Urea, 1% NP-40, 20 mM HEPES pH 7.5, 7.5 mM MgCl₂, 0.2 mM EDTA, 1 mM DTT), complemented with 20 units SUPERaseIn RNase Inhibitor (Life Technologies # AM2694) and 1× cOmplete EDTA-free Protease Inhibitor Cocktail (Roche cat nr. 11873580001). The samples were vortexed for 30 s and centrifuged at 12 500 G at 4 °C for 30 min. The chromatin pellet (Supplementary Fig. 1A) was washed once in ice cold 50 mM Tris-HCl (pH 7.5) and stored in chromatin storage buffer (50 mM Tris-HCl pH 8.0, 25% glycerol, 5 mM MgAc₂, 0.1 mM EDTA, 5 mM DTT, 20 units SUPERaseIn RNase Inhibitor) at −80 °C.

### Chromatin fragmentation

Chromatin from 30 million cells was sonicated for 5 min at 4 °C using Bioruptor (Diagenode) with 30 s on / 30 s off intervals and high throughput settings. Per each sample of 30 million K562 cells, sonicated chromatin from 250 000 MEFs was added as spike-in material. The same batch of spike-in was utilized for all the samples. Given the smaller genome size of mouse ($2.5 \times 10^9$ bp) compared to human ($2.9 \times 10^9$ bp) and the hypotriploid genome of K562 cells (Naumann et al., 2001), the spike-in was estimated to corresponds to 0.5% of total chromatin per run-on reaction. The sonicated chromatin was fragmented to <80 bp with high concentration RNase-free DNase I for 5 min at 37 °C water bath [100-300 units of DNase I and 1x DNase I buffer (ThermoFisher, cat.nr. EN0523), 1 mM DTT, 1×cOmplete EDTA-free Protease Inhibitor Cocktail (Roche cat nr. 11873580001) and 20 units SUPERaseIn RNase Inhibitor].

### Run-on reaction

2x run-on mix [10 mM Tris-Cl pH 8.0, 5 mM MgCl₂, 1 mM DTT, 300 mM KCl, 50 µM biotin-11-A/C/G/UTP (Perkin Elmer cat nrs. NEL544001EA, NEL542001EA, NEL545001EA, NEL543001EA), 40 units SUPERaseIn RNase Inhibitor, 1% sarcosyl] was prepared and pre-warmed at 37 °C water bath. Directly after the 5-min DNase I treatment, pre-warmed 2x run-on mix was added, and the run-on reaction conducted by maintaining the samples at 37 °C for an additional 5 min.

### Immunoprecipitation of Pol II complexes

The run-on reaction was terminated by diluting the chromatin in ice-cold chromatin immunoprecipitation buffer [final concentration 150 mM NaCl, 20 mM Tris-Cl pH 8.0, 1% Triton X-100, 1x proteinase inhibitors, 1x phosphatase inhibitors (phosSTOP, Roche cat. nt. 4906837001), and 20 units of SUPERaseIn RNase inhibitors]. The run-on chromatin was pre-cleared with pre-washed protein G beads (Invitrogen Dynabeads, cat. nr. 10004D or GE Healthcare Sepharose, cat nr. GE17-0618-01), and each sample divided into distinct tubes for antibody pulldowns (see Fig. 1a). The antibody pulldowns were conducted at 4 °C using 5 µg of antibody, pre-bound to 20 µL washed protein G beads, under end-to-end rotation for 1–4 h. The antibodies used in the pulldown experiments (Supplementary Fig. 1B) were raised against unphosphorylated Pol II CTD (Abcam, 8WG16, RRID: AB_2268549), serine-2-phosphorylated Pol II CTD (Millipore, 04-1571, lot number 3122213, 3E10, RRID: AB_2631403; MBL International, Clone MABI0602, RRID: AB_2747403), serine-5-phosphorylated Pol II CTD (Millipore, 04-1572, lot number 3226526, 3E8. RRID: AB_2801296; MBL International, Clone MABI0603, RRID: AB_2728736), and serine-7-phosphorylated CTD (Millipore 04-1570, lot number 1951949, 4E12, RRID: AB_2801298). The negative IgG control antibody was from Santa Cruz (sc-2027, RRID: AB_737197). The beads were washed with ChIP wash buffer A (0.1% SDS, 0.1% Triton X-100, 150 mM NaCl, 2 mM EDTA pH 8.0, 20 mM Tris-HCl pH 8.0), ChIP wash buffer B (0.1% SDS, 0.1% Triton X-100, 500 mM NaCl, 2 mM EDTA pH 8.0, 20 mM Tris-HCl) and ChIP wash buffer C (2 mM EDTA pH 8.0, 20 mM Tris-HCl pH 8.0, 10% glycerol), and diluted in TE-buffer (10 mM Tris-HCl pH 8.0, 1 mM EDTA pH 8.0). NoAb (PRO-seq) control sample was maintained rotating at 4 °C without being subjected to antibody pulldowns or antibody-bead washes.

### Isolation of nascent transcripts

The samples were diluted in TRIzol LS and chloroform, homogenized by vortexing, incubated 1 min on ice and centrifuged 18 000 G, 4 °C for 5 min. The aqueous layer, containing total RNA in the chromatin-Pol II complexes, was collected, and the RNA pelleted with EtOH using glycoblue (ThermoFisher, AM9516) as a marker of the precipitate. The RNA pellet was washed with 70% EtOH, air-dried, and diluted in RNase-free water. The RNA was base hydrolyzed with NaOH for 5 min and the base hydrolysis terminated with Tris-HCl pH 6.8. NoAb sample was passed through P-30 columns (BioRad, cat.nr. 7326232) to remove free biotinylated nucleotides. The nascent RNAs, containing biotin-nt at the 3′-end, were purified from the non-nascent RNAs using Streptavidin Dynabeads (Invitrogen, MyOne C1, cat. nr. 65002). The bead-nascent-RNA-complexes were washed with PRO-seq wash buffers 1 (50 mM Tri-Cl pH 7.4, 2 M NaCl, 0.5% Triton X-100), 2 (10 mM Tris-Cl pH 7.4, 300 mM NaCl, 0.1% Triton X-100) and 3 (5 mM Tris-Cl pH 7.4, 0.1), after which the RNA was isolated with TRIzol and chloroform, and precipitated with EtOH.

### 3′-barcoding and recombining PRO-IP-seq samples

The nascent RNA pellet was diluted in barcoded 3′-adapters, and intermolecular interactions were disrupted at 65 °C for 20 sec. The barcoded adapters were ligated to the nascent transcripts over night at 25 °C using T4 RNA Ligase I (NEB, M0204L) in 10 µl reaction volume and 5 µM final 3′-adapter concentration. The sample-specific barcoded 3′-adapters used in this study contain a constant G at the ligation site (5′-end of the adapter), followed by a 6-nt in-line barcode, and an inverted T at the 3′-end to prevent adapter concatemers[53]. An example 3′-adapter is shown below, indicating the sample-specific in-line barcode inside the square brackets.

rG[rArUrCrArCrG]rCrGrArUrGrUrGrArUrCrGrUrCrGrGrArCrUrGr
UrArGrArArCrUrCrUrGrArArC/3InvdT/

The unligated adapters were removed by binding the nascent transcripts to Streptavidin Dynabeads and washing the beads with

PRO-seq wash buffers 1 and 3. During the washes, the samples originating from a run-on reaction were combined in a single new tube (Fig. 1a).

## 5'-decapping, 5'-hydroxyrepair and 5'-adapter ligation

The 5'-decapping and 5'-hydroxyrepair were conducted on-beads. The bead-nascent-RNA-complexes were diluted in RNA 5' Pyrophosphorylase mix (NEB, M0356S) and incubated at 37 °C for 45 min. The 5'-hydroxyrepair was subsequently conducted by adding T4 polynucleotide kinase mix (NEB, M0201S) and maintaining the samples at 37 °C for additional 45 min. The bead-nascent-RNA-complexes were washed with PRO-seq buffers 1 and 3. The nascent transcripts were isolated from the beads with TRIzol and chloroform, and precipitated with EtOH. The air-dried RNA was diluted in 5'-adapter, and the sample incubated at 65 °C for 20 s. The UMI-containing 5'-adapters were ligated to the nascent transcripts with T4 RNA Ligase I over night at 25 °C in 20 μl reaction volume and 2.5 μM final 5'-adapter concentration. The 5'-adapter contained a constant C at the ligation site (3'-end of the adapter), an adjacent 6-nt UMI, and an inverted T at the 5'-end to prevent adapter concatemers[53]. The sequence of the 5'-adapter is shown below, indicating the UMI bolded inside the square brackets.

/5InvddT/CCTTGGCACCCGAGAATTCCA**[NrNrNrNrNrN]**rC

The unligated adapters were removed by binding the nascent transcripts to Streptavidin Dynabeads and washing the bead-nascent-RNA-complexes with PRO-seq wash buffers 1, 2 and 3. The nascent transcripts were isolated with TRIzol and chloroform, and precipitated with EtOH.

## Reverse-transcription and amplification

The pellet of nascent transcripts was diluted in a mix of RP1-primer and dNTPs, and intermolecular interactions were disrupted at 65 °C for 20 s. Reverse transcription was performed with Super Script III reverse transcriptase (ThermoFisher, 18080044) in 25 μl reaction volume containing 2.5 μM RP1-primer, 625 μM dNTPs and 20 units SUPERaseIn RNA inhibitors. The reverse-transcription was conducted using the following settings: 15 min at 45 °C, 40 min at 50 °C, 10 min at 55 °C, 15 min at 70 °C. The reverse-transcribed samples were test-amplified (as described in 44) and visualized on a 6% PAGE. The samples were then amplified in a total of 12 cycles using Phusion Polymerase (in-house) in 1x HF buffer (NEB, M0530S), 1 M betaine, 0.25 mM dNTPs, 0.2 μM RP1 primer and 0.2 μM RPI-n index primer. The amplified samples were purified with 1.6x Ampure XP beads (Beckman Coulter, cat. nr. A63880). The samples were analyzed with BioAnalyzer and sequenced with Illumina HiSeq2500 (Cornell University, NY USA) or NovoSeq6000 (NovoGene Inc, CA, USA) using either SE75 or PE150 settings. RP1 and RPI-n primers are Illumina small RNA TruSeq design (Oligonucleotide sequences © 2015 Illumina, Inc. All rights reserved).

## Computational analyses of PRO-IP-seq datasets

The computational analyses of PRO-IP-seq largely follows the established analyses pipelines of PRO-seq[53,76]. Here, the PRO-IP-seq samples originating from one run-on reaction were barcoded with an in-line hexamer in the 3'-adapter[53]. In the computational analyses, reads in a pool of samples were first separated using fastx barcode splitter (http://hannonlab.cshl.edu/fastx_toolkit/), and when paired-end sequencing was used, correct pairing of reads ensured with fastq_pair (https://github.com/linsalrob/fastq-pair). The adapters sequences (TGGAATTCTCGGGTGCCAAGGAACTCCAGTCAC in Read 1; GATCGTCGGACTGTAGAACTCTGAACGTGTAGATCTCGGTGGTCGCC GTATCATT in read 2) were then removed and the 6-nt UMI in the 5'-adapter was used to collapse PCR duplicates into a single read using fastp[77]. The 7-nucleotide sequences of barcode+C and UMI + C were removed and the Read 1 reverse complemented with fastx toolkit. To

avoid mapping human reads to the mouse spike-in genome, we first used high sequencing depth PRO-seq data (80 million uniquely mapping reds) that did not contain foreign DNA[24], and aligned it to mm10 using Bowtie2[78]. Regions of mm10 where cross-mapping from human nascent transcriptome occurred were masked with maskFasta (bedtools package[79]). Then, PRO-IP-seq datasets were mapped to the masked mm10 spike-in genome, and uniquely mapped reads used to calculate $nf_{spike-in}$. The PRO-IP-seq reads were subsequently mapped to the human genome (hg19) with Bowtie2. Mapped reads were processed from bam files to bed, bedgraph and bigwig files[76] using samtools[80], bedtools[79], and bedGraphToBigWig (https://www.encodeproject.org/software/bedgraphtobigwig/). The complete PRO-IP-seq datasets with raw (fastq) and processed (bigWig) files are available via Gene Expression Omnibus database (https://www.ncbi.nlm.nih.gov/geo/) with accession code GSE200269.

## Normalization of PRO-IP-seq datasets

The samples were normalized using two distinct strategies. First, we generated sequencing-depth-normalized density profiles of the NHS samples, producing patterns of engaged Pol II (Supplementary Fig. 2A) that agree with most of the previously reported results from distinct laboratories (reviewed in ref. 5). This RPM normalization, however, assumes samples to have the same level of total transcription, and it skews analyses when transcriptional activity changes genome-wide, occurring for example during heat shock[23–25,53]. We, therefore, refined the normalization strategy by deriving normalization factors (nfs) from reads mapping to spike-in genome[81] ($nf_{spike-in}$), and to ends of long (>150 kb) genes ($nf_{longGE}$) where heat-triggered changes in transcription had not proceeded during the 30-min heat shock[24,25,53]. The $nf_{spike-in}$ was counted against the unphosphorylated CTD from the same run-on reaction ($nf_{spike-in}$ = spike-inCount$_{sample}$/spike-inCount$_{unPhCTD}$). HS30 sample pool was then adjusted against the NHS pool using the ends of long genes (endCount$_{unPhCTD\_HS30}$/endCount$_{unPhCTD\_NHS}$). In this strategy, a designed control sample (here unPhCTD in NHS condition) gets multiplied with 1. Finally, all the samples are RPM normalized using sequencing depth from the control sample (Supplementary Fig. 2B). This nf-corrected control RPM (nf-cRPM) adjusts samples within an experiment using extrinsic (spike-in genome) and/or intrinsic (unchanged regions) normalization strategies, and then brings transcription profiles in different run-on experiments to comparable y-scale using sequencing depth using the control condition (Supplementary Fig. 2C, D). Importantly, nf-cRPM assumes equal pull-down of spike-in material. Even this strategy is likely to even-out differences between antibodies within a condition, but it does not force the CTD modifications to the same transcriptional range as occurs in traditional RPM normalization.

## Visualization of genomic data

PRO-IP-seq (this study, GSE200269), mNET-seq[47,48] (GSE60358; GSE106881) and POINT-seq[82] (GSE159326) datasets were visualized using an in-house tool (Hojoong Kwak, Cornell University, Ithaca, USA) and Integrative Genomics Viewer[83]. The mNET-seq and POINT-seq data were downloaded as normalized bigWig files and visualized using the genome version indicated at the site of download (hg19 or hg38). The scale of y-axis in PRO-IP-seq is comparable to sequencing depth-normalized control sample (nf-cRPM explained above). In each browser image, the scale is indicated in the above most track, and is the same and linear for all tracks in the browser image. The signal on the plus strand is shown on a positive scale, the signal on the minus strand is shown on a negative scale. The Hi-C data[84] was visualized using 3D Genome Browser[85] at 25 kb resolution and the Pol II ChIA-PET data[86] (GSE33664) was downloaded in hg19 mapped bed-format from ucsc genome browser (https://genome.ucsc.edu/cgi-bin/hgGateway) and visualized as connected blocks with IGV.

## Quantification of Pol II densities at genomic loci

The composite profiles of factor intensities from ChIP-seq, enzymatic footprints from MNase-seq and DNase I-seq, and engaged Pol II complexes from PRO-IP-seq, were generated using bigWig package (https://github.com/andrelmartins/bigWig/). The normalized count of engaged Pol II was queried at defined genomic loci, using a query window of 1 nt, 5 nts, or a 1/50th fraction of the gene body. The queried genomic sites and query windows are indicated in respective figure and figure legend. The average intensities are show with solid lines, the shaded areas display 12.5–87.5% confidence interval.

## Comparison of phosphorylated Pol II CTD against unphosphorylated CTD

In this study, the antibody raised against unphosphorylated CTD of Pol II (8WG16) was selected for normalization baseline (obtaining nf 1). This antibody was used across all the replicates (Supplementary Fig. 1B), and it was preferred over PRO-seq (NoAb) control due to PRO-seq reporting all nascent transcripts, not only transcription by Pol II. Worth noting is that the protein levels of Pol II, detected with 8WG16 antibody, remain unchanged upon and after heat shock exposures in K562 cells and MEFs[53]. The comparison of phosphorylated to unphosphorylated CTD was conducted by first counting the density of engaged Pol II bearing the respective modifications at nucleotide resolution (from normalized bigwig files). Then, the count of engaged Pol II with a given CTD modification was divided with the count of unphosphorylated Pol II CTD at each nucleotide. This ratio was set to log2-scale to center the signal to 0 (see Fig. 1d–f).

## Quantification of transcription and mRNA expression

Transcriptional activity for each gene was measured as previously described[76] using gene body as the region marking productive elongation. The normalized gene body count of engaged Pol II molecules was divided with gene body length multiplied with 1000. The resulting gene body RPK (gbRPK) reports the density of Pol II along the gene body. To compare gene transcription in PRO-IP-seq *versus* PRO-seq data, we combined the normalized gbRPK signals from different antibodies in each condition (sum CTD antibodies = unPh + Ser2Ph + Ser5Ph + Ser7Ph). The mRNA expression of respective gene was counted from existing mRNA-seq data in K562 cells[87], remapped to hg19 as previously described[88] (Burchfiel et al., 2021).

## Identification of the Transcription Start Nucleotide (TSN)

The exact transcription start nucleotide was called using the 5'-ends of PRO(-IP)-seq reads as previously described[53]. In brief, the 5'-nucleotide from each read was retained and reported as a bed file. Next, the region −100–+400 from annotated TSS was queried to find the genomic coordinate with highest count of 5'-nts. The +1 nt was identified from each PRO-IP-seq dataset reported here. Since the +1 nts were highly similar between the PRO-IP-seq datasets (Supplementary Fig. 4), we used +1 nt identified from unphosphorylated CTD under NHS as a representative of the +1 nt (TSN) throughout the study.

## Identification of the pause nucleotide and pause site distributions

The exact pause nucleotide was called from the 3'-ends of PRO(-IP)-seq reads as previously described[53]. In brief, the 3'-nucleotide from each read was retained in a bed file. Next, the region −100 to +400 from annotated TSS was queried to find the genomic coordinate with highest count of 3'-nts. The distance of the pause nucleotide from the +1 nt was counted for each gene and plotted as a histogram for the distinct PRO-IP-seq datasets.

## Western blotting

Analyses of Pol II protein levels were conducted as previously described[53] with minor modifications: Cells were lysed in buffer C (25% glycerol, 20 mM HEPES pH 7.4, 1.5 mM $MgCl_2$, 0.42 M NaCl, 0.2 mM EDTA, 0.5 mM PMSF, 0.5 mM DTT), and protein concentration in the soluble fraction was measured using Qubit. 20 μg of total soluble protein was loaded into SDS-PAGE gel and transferred to nitrocellulose membrane (Protran nitrocellulose; Schleicher & Schuell). The membranes were incubated with primary antibodies against Pol II RPB1, either the N-terminal domain (polyclonal, Santa Cruz, sc899, RRID: AB_632359), used in 1:200 dilution, or the CTD (monoclonal antibodies: serine-2-phosphorylated, Millipore, 04-1571, lot number 3122213, 3E10, RRID: AB_2631403; serine-5-phosphorylated, Millipore, 04-1572, lot number 3226526, 3E8. RRID: AB_2801296; and serine-7-phosphorylated, Millipore 04-1570, lot number 1951949, 4E12, RRID: AB_2801298), each used in 1:1000 dilution. β-tubulin (Abcam, ab6046, RRID: AB_2210370) in 1:5000 dilution was visualized as a control for loading. The secondary antibodies were HRP conjugated from Pierce (goat anti-rabbit 1858415, lot # HJ108849; goat anti-mouse 1858413, lot HA102346 and HJ108266) or Invitrogen (anti-rat A10549), used in dilutions recommended by manufacture, and the blots were developed using an enhanced chemiluminescence method (ECL kit; GE Healthcare). The original scans of the films are available in Mendeley (https://data.mendeley.com/datasets/s44mkg6jmb/1).

## Reporting summary

Further information on research design is available in the Nature Portfolio Reporting Summary linked to this article.

## Data availability

The complete raw datasets and normalized density profiles generated (GSE200269) and used (GSE89382; GSE60358; GSE106881; GSE159326) in this study can be accessed via GEO (http://www.ncbi.nlm.nih.gov/geo) and ENCODE (https://www.encodeproject.org) databases. Source data are provided with this paper.

## Code availability

Pipelines for mapping the raw data and generating density profiles are available in GitHub (https://github.com/Vihervaara/PRO-IP-seq). Custom made scripts for image generation are available from the corresponding author upon request.

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

## Acknowledgements

We thank the members of Vihervaara and Lis laboratories for valuable advice and discussions. Alex Greenstone is gratefully acknowledged for the expert assistance with antibody optimization. This work was financially supported by Science for Life Laboratory (A.V.), Swedish Research Council (A.V.), Academy of Finland (A.V.), National Science Foundation Graduate Research Fellowship Program grants DGE-1650441 and DGE-21389899 (P.V.), and NIH grants R01-GM25232 (J.T.L) and RM1-GM139738 (J.T.L.). The content is solely the responsibility of the authors and does not necessarily represent the official views of the NIH. Any opinions, findings, and conclusions or recommendations expressed in this material are those of the author(s) and do not necessarily reflect the views of the National Science Foundation or National Institutes of Health.

## Author contributions

A.V. and J.T.L. conceived and designed the study. A.V. generated the PRO-IP-seq protocol, which was optimized by P.V. A.V. and S.H. conducted the computational analyses. All authors interpreted the results and wrote the manuscript.

## Funding

## Competing interests

The authors declare no competing interests.
