## [Peer Review File · Nature Communications]

PRO-IP-seq Tracks Molecular Modifications of Engaged Pol II Complexes at Nucleotide ResolutionEditorial Note: Parts of this Peer Review File have been redacted as indicated to remove third-party material where no permission to publish could be obtained.

REVIEWER COMMENTS

Reviewer #1 (Remarks to the Author):

Here Vihervaara, Versluis, and Lis, present an enhanced version of PRO-seq called PRO-IP-seq that allows the authors to simultaneously measure nascent transcription and CTD phosphorylation status. They use this method to show that during heat shock, phosphorylation patterns change in promoter proximal regions (no phosphorylation to phosphorylation at Ser2 and Ser5). They also observe high levels of Ser7 phosphorylation after RNA polymerase II escapes from promoter proximal pausing. The work is well done and provides a method to investigate how post-translational modifications or factor association are associated with distinct transcriptional states. I have a few minor comments that the authors should address prior to publication.

Line 116: The authors state that the run-on reaction was performed with spike-ins. It is unclear how these spike-ins are used because it seems like they would be lost during the subsequent IP steps. The authors should clarify this in the main text in addition to what they have provided in the methods section.

Lines 451-453: The authors could also mention that mNET-seq has also been modified to do similar things. <https://doi.org/10.1016/j.crmeth.2022.100368>

Line 542, 626- The authors mention YWG16- I think they intend to cite 8WG16

Lines 627-629- In addition to the manufacturer numbers, the authors should provide the antibody names from Chapman et al., 2007. (e.g., 3E8 for Ser5, 4E12 for Ser7). The authors should also cite this paper: DOI: 10.1126/science.1145977 since it shows the specificity differences for the different CTD antibodies and lends support to their claim that they are looking at specific phosphorylation states.

Reviewer #2 (Remarks to the Author):

Here, Vihervaara, Versluis and Lis describe a modification of the Pro-seq protocol to analyse CTD phosphorylation patterns of POL2 genome-wide in human cells. After isolation and fragmentation of chromatin and the run-on reaction, which allows the engaged Pol II machinery to incorporate a labelled nucleotide, different antibodies detecting the phosphorylation status of POL II were used for immunoprecipitation. The distribution of the nascent transcripts, and thus of the Pol II complexes with the modifications analysed, was then determined by Illumina sequencing after 3'-adapter ligation and reverse transcription. The authors describe the distribution of pS2, pS5 and pS7 Pol II compared to unphosphorylated Pol II (Figure 1). They describe a shift in phosphorylation during pausing with predominantly unphosphorylated Pol II in the 'early pause site' and a rapid increase in pS5 and pS2 thereafter, which they call the 'late pause site' (Figure 2). They then focused on pS7, which peaks at the 5' end of genes between the pause site and +1000 nucleotides (Figure 3). The authors then compared the Po-IP-seq profiles of unperturbed and heat-shocked cells and found distinct redistributions of heat-induced genes compared to heat-repressed genes (Figure 5, 6). While this method can measure Pol II distributions according to their phosphorylation pattern at single nucleotide resolution, it's advantage over similar methods mapping POL II distributions such as mNET-seq is not huge. The paper is well written and easy to understand, but lacks a clear mechanistic take-home message. I see three main limitations.

1.: The authors have not made the necessary effort to compare their results with the literature and to point out what they have found new and what fits (or does not fit) with previous observations. In particular, I miss comparisons with mNET-seq papers, especially the seminal studies from the

Proudfoot lab (Nojima et al., 2015). Do the authors see pS3 peaks at exon-intron boundaries?

2. The most interesting observations of this paper are (i) the transient pS7 peaks between the pause site and +1000 and their predictive power for transcriptional activity, and the fact that pS2/5 is not sufficient to drive transcriptional elongation at heat-repressed genes. However, these statements are based on the analysis of a single replicate. ALL key statements of this manuscript should be based on all replicates and triplicate analyses should be made available to the reader, e.g. by presenting one replicate in the main text and the same analysis of the other replicates in the supplement. For the analyses shown in Figures 1c; 2c; 3c; 6C, please specifically show the two missing replicates. Please also show the IgG control in the main figures.

- 3. The changes in chromatin architecture upon heat shock as measured by Hi-C (Figure 4) are largely unrelated to the Pro-IP protocol and data sets of the study.

Minor points:

- 4e, quantification of biological replicates.

- Scale, sequencing depth

- In most of the panels showing single genes (e.g. 1B, 3A, B, ...), the signals are difficult to see.

Either the scaling is not ideal or the libraries are under-sequenced. Why are there gaps in some genes, e.g. in the middle of the RPL15 gene (Figure 1B) and the ACTB gene (Figure 2b)?

- Figure 4e and the corresponding text (line 383-385) do not correlate. pS2 increases after 60 minutes, not 30 minutes, and pS5 does not increase much at all.

Point-by-point response:

We thank the Reviewers for carefully reading our manuscript and their helpful advice in its improving. We have amended the manuscript according to the Reviewers' comments. Below, please find our point-by-point response to all the concerns raised. The Reviewers' comments are indicated in blue, followed by our response. The changes made to the manuscript text are shown in red in this response as well as in the amended manuscript file.

=====

REVIEWER COMMENTS

Reviewer #1 (Remarks to the Author):

Here Vihervaara, Versluis, and Lis, present an enhanced version of PRO-seq called PRO-IP-seq that allows the authors to simultaneously measure nascent transcription and CTD phosphorylation status. They use this method to show that during heat shock, phosphorylation patterns change in promoter proximal regions (no phosphorylation to phosphorylation at Ser2 and Ser5). They also observe high levels of Ser7 phosphorylation after RNA polymerase II escapes from promoter proximal pausing. The work is well done and provides a method to investigate how post-translational modifications or factor association are associated with distinct transcriptional states. I have a few minor comments that the authors should address prior to publication.

Line 116: The authors state that the run-on reaction was performed with spike-ins. It is unclear how these spike-ins are used because it seems like they would be lost during the subsequent IP steps. The authors should clarify this in the main text in addition to what they have provided in the methods section.

Thank you for raising this point. We chose mouse spike-in material to ensure that the utilized antibodies can recognize Pol II CTD in the sample (human) and spike-in (mouse) alike. In essence, both sample and spike-in material are carried through the protocol in an equal manner. We now explain this aspect in the main manuscript text (lines 109-115 in the amended manuscript): **“To obtain an external normalization control that follows through the protocol, we spiked-in mouse chromatin to each sample before the run-on reaction. This ensures that throughout the protocol, human and mouse chromatin are treated identically, including the biotinylation of 3'-ends of nascent RNAs during run-on reaction, immunoprecipitation of transcription complexes using antibodies that recognize both human and mouse Pol II CTD, isolation of biotinylated RNAs with streptavidin, and the subsequent library preparation of the nascent RNAs.”**

Lines 451-453: The authors could also mention that mNET-seq has also been modified to do similar things. <https://doi.org/10.1016/j.crmeth.2022.100368>

We added a direct comparison of PRO-IP-seq to mNET-seq as a separate paragraph (see new Figures 2 and S6-7, and lines 192-214 of the amended manuscript):

“PRO-IP-seq is highly sensitive and specific for nascent RNAs

NET-seq (Churchman and Weissman, 2011) and its mammalian adaptation (Nojima *et al.*, 2015) have enabled seminal findings in Pol II CTD phosphorylation and transcriptional regulation (reviewed in Zaborovska *et al.*, 2016; Harlen and Churchman, 2017). Overall, PRO-IP-seq identified modifications at engaged Pol II (Figure 1B-E) correlated well with previously reported Pol II phosphorylations, deduced from mNET-seq and ChIP-seq data (reviewed in Zaborovska *et al.*, 2016; Harlen and Churchman, 2017). However, mNET-seq selects Pol II associated RNAs, not all of which are nascent, giving rise to various signal peaks that are not present in run-on based assays (reviewed in Wissink *et*

al., 2019). Accordingly, PRO-IP-seq did not contain spurious spikes, which are present in various mNET-seq datasets, with or without the use of empigen (Figures 2 and S6-7). As an example, PRO-IP-seq did not enrich for engaged Pol II bearing serine-5-phosphorylation at exon-intron boundaries (Figure 2A-C, S6F and S7A), initially reported in mNET-seq (Nojima *et al.*, 2015) and later identified to be non-nascent RNA from the spliceosome (Nojima *et al.*, 2018). Of note is that the spurious spikes in mNET-seq data extended from gene bodies (Figures 2D-E, S6A-B, S7C) to promoter-proximal regions (Figures 2F, 2H, S6C and S7D), obscuring the detection of Pol II initiation and pausing. In PRO-IP-seq, the double enrichment for a modification of interest and nascent RNA allowed identification of the precise initiation coordinate and Pol II pause sites. This is demonstrated by the single Pol II pause-region (Figures 2A, 2C, 2G, S6F-G and S7A-B) and precise +1 nts (Figures S4 and S6G) in PRO-IP-seq data, compared to the several potential pause-regions (Figures 2D, 2F, S6A-C and S7C-D) and initiating nucleotides (Figure S6D-E) in mNET-seq assays. We conclude that nucleotide-resolution identification of the initiation and active sites of transcription from affinity-purified nascent RNAs enables tracking Pol II CTD modifications at unprecedented resolution and sensitivity (Figures 1-2).”

Please note that comparison to mNET was also requested by Reviewer 2, and an elaborated discussion on PRO-IP-seq *versus* mNET-seq is provided as a response to the major point 1 of Reviewer 2.

Line 542, 626- The authors mention YWG16- I think they intend to cite 8WG16

Yes, thank you! This is now corrected throughout the manuscript.

Lines 627-629- In addition to the manufacturer numbers, the authors should provide the antibody names from Chapman *et al.*, 2007. (e.g., 3E8 for Ser5, 4E12 for Ser7). The authors should also cite this paper: DOI: 10.1126/science.1145977 since it shows the specificity differences for the different CTD antibodies and lends support to their claim that they are looking at specific phosphorylation states.

Thank you for the helpful advice for specifying the antibodies and indicating the reference. We added the above mentioned specifications, as well as the Research Resource IDentification (RRID) code (<https://www.rrids.org>) for every antibody used in this study (lines 547-552 for PRO-IP-seq, and lines 734-737 for WB, in the amended manuscript).

We also cite Chapman *et al.*, 2017 in the main text for verified specificity of Pol II CTD targeting antibodies (lines 119-120 in the amended manuscript): **The antibodies were previously verified to be specific against the distinct Pol II CTD modifications (Chapman *et al.*, 2007).**

Reviewer #2 (Remarks to the Author):

Here, Vihervaara, Versluis and Lis describe a modification of the Pro-seq protocol to analyse CTD phosphorylation patterns of POL2 genome-wide in human cells. After isolation and fragmentation of chromatin and the run-on reaction, which allows the engaged Pol II machinery to incorporate a labelled nucleotide, different antibodies detecting the phosphorylation status of POL II were used for immunoprecipitation. The distribution of the nascent transcripts, and thus of the Pol II complexes with the modifications analysed, was then determined by Illumina sequencing after 3'-adapter ligation and reverse transcription. The authors describe the distribution of pS2, pS5 and pS7 Pol II compared to unphosphorylated Pol II (Figure 1). They describe a shift in phosphorylation during pausing with predominantly unphosphorylated Pol II in the 'early pause site' and a rapid increase in pS5 and pS2 thereafter, which they call the 'late pause site' (Figure 2). They then focused on pS7, which peaks at the 5' end of genes between the pause site and +1000 nucleotides (Figure 3). The authors then compared the Po-IP-seq profiles of unperturbed and heat-shocked cells and found distinct

redistributions of heat-induced genes compared to heat-repressed genes (Figure 5, 6). While this method can measure Pol II distributions according to their phosphorylation pattern at single nucleotide resolution, its advantage over similar methods mapping POL II distributions such as mNET-seq is not huge. The paper is well written and easy to understand, but lacks a clear mechanistic take-home message. I see three main limitations.

1.: The authors have not made the necessary effort to compare their results with the literature and to point out what they have found new and what fits (or does not fit) with previous observations. In particular, I miss comparisons with mNET-seq papers, especially the seminal studies from the Proudfoot lab (Nojima *et al.*, 2015). Do the authors see pS3 peaks at exon-intron boundaries?

We do appreciate that direct comparison to mNET-seq is required, and added a new paragraph with three new Figures (2 and S6-7) to compare PRO-IP-seq versus mNET-seq.

Please note that in 2015 Proudfoot lab (Nojima *et al.*, 2015, Cell) published a mammalian version of Native Elongation Sequencing (NET-seq), adapting the original publication in yeast by Weissman lab (Churchman and Weissman, 2011, Nature). The major difference between PRO-IP-seq and mNET-seq is that PRO-IP-seq selects nascent RNAs while mNET-seq isolates Polymerase-associated RNAs. Some of the Polymerase-associated RNAs in mNET-seq are not nascent, yielding various 'spikes' in the mNET-seq data that we do not detect in nascent RNA-sequencing assays (GRO-seq, PRO-seq, CHRO-seq, PRO-IP-seq). These non-nascent RNAs include, for example, U6 RNA that was initially reported to enrich with serine-5-phosphorylated Pol II (Nojima *et al.*, 2015, Cell) but later corrected by the same group to consist of non-nascent transcripts originating from the splicing machinery (Nojima *et al.*, 2018, Mol. Cell). Even the newest POINT and POINT5 -techniques, which added a 3% emgigen wash to mNET-seq (Sousa-Luis *et al.*, 2021, Mol Cell), contains signal that to us looks non-nascent and shows blurred signal for the +1nt analyses. Consequently, and as expected, we do not detect increased serine-5-phosphorylation, or any other CTD modification, to enrich at the intron-exon-boundaries. Downloading and visualization of the mNET-seq and POINT-seq data is described in Materials and Methods (see lines 673-678) of the amended manuscript. Below, please find the paragraph we added for direct PRO-IP-seq *versus* mNET-seq comparison (lines 192-214 of the amended manuscript) as well as one of the added figures (Supplemental Figure 7):

“PRO-IP-seq is highly sensitive and specific for nascent RNAs

NET-seq (Churchman and Weissman, 2011) and its mammalian adaptation (Nojima *et al.*, 2015) have enabled seminal findings in Pol II CTD phosphorylation and transcriptional regulation (reviewed in Zaborovska *et al.*, 2016; Harlen and Churchman, 2017). Overall, PRO-IP-seq identified modifications at engaged Pol II (Figure 1B-E) correlated well with previously reported Pol II phosphorylations, deduced from mNET-seq and ChIP-seq data (reviewed in Zaborovska *et al.*, 2016; Harlen and Churchman, 2017). However, mNET-seq selects Pol II associated RNAs, not all of which are nascent, giving rise to various signal peaks that are not present in run-on based assays (reviewed in Wissink *et al.*, 2019). Accordingly, PRO-IP-seq did not contain spurious spikes, which are present in various mNET-seq datasets, with or without the use of emgigen (Figures 2 and S6-7). As an example, PRO-IP-seq did not enrich for engaged Pol II bearing serine-5-phosphorylation at exon-intron boundaries (Figure 2A-C, S6F and S7A), initially reported in mNET-seq (Nojima *et al.*, 2015) and later identified to be non-nascent RNA from the spliceosome (Nojima *et al.*, 2018). Of note is that the spurious spikes in mNET-seq data extended from gene bodies (Figures 2D-E, S6A-B, S7C) to promoter-proximal regions (Figures 2F, 2H, S6C and S7D), obscuring the detection of Pol II initiation and pausing. In PRO-IP-seq, the double enrichment for a modification of interest and nascent RNA allowed identification of the precise initiation coordinate and Pol II pause sites. This is demonstrated by the single Pol II pause-region (Figures 2A, 2C, 2G, S6F-G and S7A-B) and precise +1 nts (Figures S4 and S6G) in PRO-IP-seq data, compared to the several potential pause-regions (Figures 2D, 2F, S6A-C and S7C-D) and initiating nucleotides (Figure S6D-E) in mNET-seq assays. We conclude that nucleotide-resolution identification of the initiation and active sites of transcription from affinity-purified nascent RNAs enables tracking Pol II CTD modifications at unprecedented resolution and sensitivity (Figures 1-2).”

Supplemental Figure 7

Figure S7. PRO-IP-seq is free from contaminating non-nascent RNAs. A-B) PRO-IP-seq profiles of engaged Pol II bearing distinct CTD phosphorylations across *GRIPAP1* gene. **A** shows the Pol II CTD phosphorylations across the gene, **B** zooms into the single Pol II pause-region and identifies the pause-coordinates. E: early Pol II pause (+18-30nt); L: late Pol II pause (+31-60nt). See also Figure 3 for Pol II pause coordinate identification. **C-D)** mNET-seq profiles of distinct Pol II phosphorylations across the *GRIPAP1* gene (**C**) and its promoter-proximal region (**D**). The mNET-seq data was obtained as normalised bigWig files from Nojima *et al.*, 2015. The orange arrow indicates the +1nt identified from the 5'-ends of Pol II unPhCTD PRO-IP-seq reads and is the same as indicated in Figure 3.

New Supplemental Figure 7 and its legend, comparing PRO-IP-seq and mNET-seq profiles at *GRIPAP1* gene. Example exon-intron boundaries are shown in the new Figure 2 and S6 in the amended manuscript. Please also note that we added tracks for replicates (see point 2 below).

We wish to emphasize we have selected fair and representative genes for PRO-IP-seq *versus* mNET-seq comparison. For example, the *SIK1* gene in the new Figure 2 and S6A-C is a published example gene in mNET-seq Nature Protocols by Proudfoot group (see *Figure 1 for Reviewers only* below). The *NDUFB9* gene in the new Figure S6D-G in the amended manuscript was published in the latest mNET-seq adaptation (POINT-seq) from Nojima and Proudfoot labs (Sousa-Luis et al., 2021, Mol Cell). To illustrate that mNET-seq 'peaks' not detected in PRO-seq or PRO-IP-seq are prevalent in various mNET-seq datasets, we included here *Figure 2 for Reviewers only*, containing a published mNET figure panel from Murphy group (Tellier et al., 2020, NAR). To back-up our claim on PRO-IP-seq gaining an unprecedented resolution, please compare e.g. the metaprofiles from mNET-seq (*Figure 3 for Reviewers only*, containing published average mNET-seq profiles from a review by Harlen and Churchman, 2017, Nat. Rev. Mol. Cell. Biol) and the respective PRO-IP-seq-data (Figures 1C-E in the amended manuscript, as well as Figure 3C also included here below).

[redacted]

Figure 1 for Reviewer's only. mNET-seq with distinct Pol II CTD antibodies, given as example 'gene profile to expect from mNET-seq' by Nojima *et al.*, 2016, Nat. Protoc. <https://www.ncbi.nlm.nih.gov/pmc/articles/PMC7641311/>.

[redacted]

Figure 2 for Reviewer's only. mNET-seq with distinct Pol II CTD antibodies published in Tellier et al., (2020, NAR, <https://www.ncbi.nlm.nih.gov/pmc/articles/PMC7641311/>).

[redacted]

Figure 3 for Reviewer's only. Average mNET-seq profiles with distinct Pol II CTD antibodies, compared against total Pol II (review by Harlan and Churchman, 2017, Nat. Rev. Mol. Cell. Biol). <https://www.nature.com/articles/nrm.2017.10>.

Figure 3C in this manuscript. Average PRO-IP-seq density of Pol II CTD phosphorylations as compared to unphosphorylated Pol II and placed into genomic context of Pol II pausing & +1 to +5 nucleosomes (Figure 3C in the amended manuscript). Please see Figure 1C-E for the whole-gene view of PRO-IP-seq average profile.

2. The most interesting observations of this paper are (i) the transient pS7 peaks between the pause site and +1000 and their predictive power for transcriptional activity, and the fact that pS2/5 is not sufficient to drive transcriptional elongation at heat-repressed genes. However, these statements are based on the analysis of a single replicate. ALL key statements of this manuscript should be based on all replicates and triplicate analyses should be made available to the reader, e.g. by presenting one replicate in the main text and the same analysis of the other replicates in the supplement. For the analyses shown in Figures 1c; 2c; 3c; 6C, please specifically show the two missing replicates. Please also show the IgG control in the main figures.

Thanks for the critical comment that we now address further to avoid any confusion by readers. Throughout the study, we have used two replicates, which is the standard in nascent RNA sequencing experiments. Additionally, we used a third replicate for serine-2 and serine-5 phosphorylated CTD to

have a direct comparison with the antibodies used in mNET-seq by Proudfoot lab. The replicates are, and have been, presented in Supplemental Figures 1B and 2. To strengthen the claims, we now demonstrate the replicate correlations in Supplemental Figure 3, including correlation plots and statistics (S3A) and browser track comparison of individual replicates (S3B also shown here below). Please note that Figures 1C, 2C (now 3C) and 3C (now 4C) were already generated using the two replicates for each sample. To clarify further how replicates were combined after verification of their adequate correlation we added the following sentence in the main manuscript text (lines 137-139 of the amended manuscript): “After ensuring high statistical correlation (Figure S3A) and highly similar replicate profiles (Figure S3B), we combined the replicates into a single sample profile (Figure 1B).”

We do appreciate that replicate correlation is crucial, and have now included browser track examples with replicates in the Supplementary Figures for all the key findings in this study: i) nf-cRPM normalization and replicate correlation (S3B, also below), ii) early and late Pol II pausing (S7A-B, also shown above in this rebuttal), iii) comparison of PRO-IP-seq to mNET-seq (2A-C, 2G, S6F-G and S7A-B), iv) heat-induced transcriptional change across genes and promoters (S10A-B), and the wave of receding Pol II along the long *MAP4K4* gene (S12, replicates for Figure 6C, now 7C, mentioned above).

New Supplemental Figure 3B showing replicate correlations for *RPL23*.

The IgG pulldowns yield a minute read count in PRO-IP-seq experiments: IgG samples provided < 100 000 uniquely mapped reads, while every Pol II CTD pulldown (generated from the same amount of starting material, largely in the same test tube) yielded > 20 million of uniquely mapped reads. We demonstrate the missing signal in IgG as the total read counts obtained (Supplemental Figure 1C) and in browser tracks (Supplemental Figure 1D, also shown here below). To highlight the missing signal in the IgG, we added a sentence to the main manuscript text (lines 128-129 of the amended manuscript):” **The IgG pulldown showed minute signal as total uniquely mapped reads (Figure S1C) and as density profiles (Figure S1D) as compared to the Pol II CTD pulldowns.**” To further improve the clarity on IgG signal in the manuscript, we added the nf-cRPM-normalized IgG browser track to Figure 1B. As discussed in the minor points below, PRO-seq and PRO-IP-seq signals are tricky to visualize in browser tracks due to high Pol II pause-signal in comparison to the signal along the gene body. To allow for more space to visualize the pause regions and gene body in the same track, we have chosen to omit the IgG tracks (and often also PRO-seq tracks), concentrating on showing the differences in the CTD phosphorylations.

Supplemental Figure 1C and D showing the minute signal in the negative IgG control, as total PRO-IP-seq reads across replicates (S1C), and as a browser track with unnormalized, replicate-combined samples (S1D). The IgG is indicated with orange arrow in this rebuttal.

New Figure 1B, now containing the nf-cRPM-normalized IgG track. Please note that the very low amount of reads in the IgG samples leads to each read being amplified in a nf-cRPM-based normalization. We, therefore, chose to omit the IgG browser tracks from the subsequent figures to allow more space for the comparison of CTD phosphorylations.

3. The changes in chromatin architecture upon heat shock as measured by Hi-C (Figure 4) are largely unrelated to the Pro-IP protocol and data sets of the study.

We have used the Hi-C and ChIA-PET data to demonstrate the large scale changes in Pol II CTD phosphorylation at connected genomic loci. We have streamlined the the text in this paragraph and aimed to improve the focus, as large-scale changes in Pol II CTD modifications, especially at interconnected TADs, have not been previously shown. The amended text is found in the lines 292-316 of the amended manuscript and altered or added text is shown in red. (Please note that Figure 4 is Figure 5 in the amended manuscript.)

Minor points:

- 4e, quantification of biological replicates.

We now provide quantification of the Western Blot results in Figure 5E (Figure 4E in the previous version). The Figure below has been added as a new panel, Figure 5F, into the amended manuscript. The figure legend of panels E and F in the new Figure 5 are as follows: “E) Representative Western

Blotting and F) quantification of total Pol II and CTD phosphorylations from three Western Blot replicates. Standard error of mean and p-values < 0.05 (student's t-test) are shown.”

Figure 5F in the amended manuscript.

- Scale, sequencing depth

Sequencing depth is shown in Supplemental Figure 1C, and as mentioned above, we added a sentence comparing the IgG and Pol II CTD sequencing depths (lines 128-129 of the amended manuscript):” **The IgG pulldown showed minute signal as total uniquely mapped reads (Figure S1C)... as compared to the Pol II CTD pulldowns.**”

We also added “deeply sequenced (Figure S1C)” to the following sentence (line 127 of the amended manuscript):

“...and the Pol II CTD pulldowns generated **were deeply sequenced (Figure S1C), yielding** nucleotide-resolution density profiles (Figure S1D) characteristic of nascent transcription.”

We have used spike-in corrected cRPM normalization which sets the y-axis to a comparable scale of sequencing depth-normalized samples. We added the following text to Materials and Methods (lines 679-681 of the amended manuscript): “**The scale of y-axis in PRO-IP-seq is comparable to sequencing depth-normalized control sample (nf-cRPM explained above). In each browser image, the scale is indicated in the above most track, and is the same and linear for all tracks in the browser image.**”

- In most of the panels showing single genes (e.g. 1B, 3A, B, ...), the signals are difficult to see. Either the scaling is not ideal or the libraries are under-sequenced. Why are there gaps in some genes, e.g. in the middle of the

RPL15 gene (Figure 1B) and the ACTB gene (Figure 2b)?

As mentioned above, PRO-seq and PRO-IP-seq profiles are challenging to visualize in browser tracks due to several fold higher Pol II pause-signal than gene body signal. When the whole pause-peak is shown, the gene body signal appears low. This is particularly challenging when multiple tracks need to be visualized in the same graph. Furthermore, we aim to show the nucleotide-resolution profiles of engaged Pol II without smoothening (and thereby blurring) the signal. We have improved the layout of the figures where possible to accommodate more room for the browser tracks. We also placed replicate comparisons in Supplementary Figures (S3B, S6A-C, S7-B, S9A-B, and S12), each of which show the key results in a larger figure than what we had room for in the main Figure.

- Figure 4e and the corresponding text (line 383-385) do not correlate. pS2 increases after 60 minutes, not 30 minutes, and pS5 does not increase much at all.

We changed the wording, stating that pS5 gained “a modest but statistically significant increase” while pS2 increase was mainly detected “upon 60 minutes” of heat stress. Please also see the added quantification of the Western Blots in the new Figure 5F. Lines 322-325 of the amended manuscript: “We found that the total cellular pool of Pol II gained a modest but statistically significant increase in phosphorylation at serine-5 of the CTD during heat stress (Figures 5E-F and S11A). Also, serine-2-phosphorylation increased during heat shock, but the increase was mainly detected upon 60 minutes of heat stress (Figures 5E-F and S11A).”

REVIEWERS' COMMENTS

Reviewer #1 (Remarks to the Author):

The authors have addressed all of my concerns.

Reviewer #2 (Remarks to the Author):

The authors have responded to my questions in detail and I endorse the publication of the revised manuscript in Nature Communications. Congratulations on a great paper.

We thank both Reviewers for their time and expert help. Their comments require no actions from our side (comments below for reference).

With kind regards,

Anniina Vihervaara, PhD
Assistant Professor
Science for Life Laboratory Fellow
Royal Institute of Technology
Department of Gene Technology
Tomtebodavägen 23A
171 65 Solna, Sweden

viher@kth.se
anniina.vihervaara@scilifelab.se
[+358 50 5463694](tel:+358505463694)

=====

REVIEWERS' COMMENTS

Reviewer #1 (Remarks to the Author):

The authors have addressed all of my concerns.

Reviewer #2 (Remarks to the Author):

The authors have responded to my questions in detail and I endorse the publication of the revised manuscript in Nature Communications. Congratulations on a great paper.

=====